# DataFreeShield: Defending Adversarial Attacks without Training Data

## Abstract

Recent advances in adversarial robustness rely on an abundant set of training data, where using external or additional datasets has become a common setting. However, due to security and privacy issues, it is more common that a pretrained model is available while the dataset is not. In such a scenario, existing methods that assume accessibility to the original data become inapplicable. For the first time, we propose a problem of learning *data-free adversarial robustness*, where given only a pretrained model, adversarial robustness should be achieved without accessing the training dataset. In our preliminary study, we identify that robustness without the original dataset is difficult to achieve, even with similar domain datasets. We tackle the task from two perspectives: surrogate dataset generation and adversarial training using the generated data. For dataset generation, we propose diversified sample synthesis, which largely enhances the diversity of synthetic samples that are known to have low coverage. For training, we propose a soft label loss that best learns robustness from noisy synthetic samples and a gradient refinement method toward smoother loss surfaces. Extensively validating methods using four datasets, we show that the proposed solution outperforms several baselines, demonstrating that the proposed method sets the first solution for the data-free robustness problem.

## 1 Introduction

Since the discovery of the adversarial examples (Goodfellow et al., 2015; Szegedy et al., 2014) and their ability to successfully fool well-trained classifiers, training a robust classifier has become an important topic of research (Schmidt et al., 2018; Athalye et al., 2018). If not properly circumvented, adversarial attacks can be a great threat to real-life applications such as self-driving automobiles and face recognition when intentionally abused.

Among many efforts made over the past few years, adversarial training (AT) (Madry et al., 2018) has become the de facto standard approach to training a robust model. AT uses adversarially perturbed examples as part of training data so that classifiers can learn to classify them as their original classes. Due to its success, many variants of AT have been proposed to further improve its effectiveness (Zhang et al., 2019; Wang et al., 2019; Zhu et al., 2022).

In the field of AT, it is commonly assumed that the original data is available for training. Going a step further, many approaches import external data from the same or similar domains to add diversity to the training samples (e.g., adding Tiny-ImageNet data to CIFAR-10), such that the trained model can have better generalization ability (Rebuffi et al., 2021; Carmon et al., 2019).

Unfortunately, the original training dataset is often not available in many real-world scenarios. While there are some public datasets available for certain domains (e.g., image classification), many real-world data are publicly unavailable due to privacy, security, or proprietary issues, with only the pretrained models available (Patashnik et al., 2021; Saharia et al., 2022; Ramesh et al., 2021). Therefore, if a user wants a pretrained model to become robust against adversarial attacks, there is currently no apparent method to do so without the original training data. However, from the attacker's side, creating an adversarial sample requires no access to the training data. This indicates that adversarial vulnerability clearly exists regardless of the accessibility of the original dataset.

In such circumstances, we define the problem of learning *data-free adversarial robustness*, where a *non-robustly* pretrained model is given and its robust version should be learned without access

to the original training data. To address the problem, we propose DataFreeShield, which creates a synthetic dataset and performs adversarial training on the synthetic dataset to obtain a robust model. Specifically, we propose a synthetic sample diversification method with dynamic synthetic loss modulation to maximize the diversity of the synthetic dataset.

Moreover, we devise a soft guided training loss that can maximize the transferability of robustness even under a severe distributional shift from synthetic-real discrepancy. Lastly, we propose a gradient refinement method GradRefine to obtain a smoother loss surface, to minimize the impact of the distribution gap between synthetic and real data. To the best of our knowledge, this is the first work that considers adversarial robustness in the absence of training data.

Our contributions are summarized as follows:

- For the first time, we formulate the problem of learning data-free adversarial robustness, which gives adversarial robustness to non-robustly pretrained models without the original datasets.
- We study critical components of data that contribute to adversarial robustness, and devise diversified sample synthesis, a novel technique to enhance the diversity of synthetic data.
- We propose a soft-guidance based training loss with a gradient refinement method to minimize the impact of distribution shift incurred from synthetic data training.
- We propose DataFreeShield, a first-ever approach that can effectively convert a pretrained model to an adversarially robust one and show that DataFreeShield achieves significantly better robustness on various datasets over baselines.

## 2 BACKGROUND

### 2.1 ADVERSARIAL ROBUSTNESS

Among many defense techniques for making DNN models robust against adversarial attacks, adversarial training (Madry et al., 2018) (AT) has been the most successful method, formulated as:

$$\min_{\theta} \frac{1}{n} \sum_{i}^{n} \max_{x_i' \in \mathcal{X}} \mathcal{L}(f_\theta(x_i'), y_i), \text{ where } \mathcal{X} = \{x_i' | \, \|x_i' - x_i\|_p \leq \epsilon\}, \tag{1}$$

where $\mathcal{L}$ is the loss function for classification (e.g., cross-entropy), $n$ is the number of training samples, and $\epsilon$ is the maximum perturbation limit. $x'$ is an arbitrary adversarial sample that is generated based on $x$ to deceive the original decision, where $p = \infty$ is a popular choice. In practice, finding the optimal solution for the inner maximization is intractable, such that known adversarial attack methods are often used. For example, PGD (Madry et al., 2018) is a widely-used method, such that

$$x^t = \Pi_\epsilon(x^{t-1} + \alpha \cdot \text{sign}\left(\nabla_x \mathcal{L}(f_\theta(x^{t-1}), y)\right)), \tag{2}$$

where $t$ is the number of iteration steps. For each step, the image is updated to maximize the target loss, then projected onto the epsilon ball, denoted by $\Pi_\epsilon$.

### 2.2 DATASET GENERATION FOR DATA-FREE LEARNING

When a model needs to be trained without the training data (i.e., data-free learning), one of the dominant approaches is to generate a surrogate dataset, found from data-free knowledge distillation (Lopes et al., 2017; Fang et al., 2019), data-free quantization (Xu et al., 2020; Choi et al., 2021; 2022), or data-free model extraction (Truong et al., 2021). With the struggle of not having real data, many works rely on a pretrained model and utilize its knowledge to recover samples from scratch. The choice for the specific synthesis loss varies from paper to paper, but the most common choice in the literature (Wang et al., 2021; Yin et al., 2020; Ghiasi et al., 2022) are as follows:

$$\mathcal{L}_{class} = \mathcal{L}_{CE}(f_\theta(x), y), \tag{3}$$

$$\mathcal{L}_{feature} = \sum_{l=1}^{L} \|\mu_l^T - \mu_l\|_2^2 + \|\sigma_l^T - \sigma_l\|_2^2, \tag{4}$$

$$\mathcal{L}_{prior} = \sum_{i,j} \|\hat{x}_{i,j+1} - \hat{x}_{i,j}\|_2^2 + \|\hat{x}_{i+1,j} - \hat{x}_{i,j}\|_2^2, \tag{5}$$

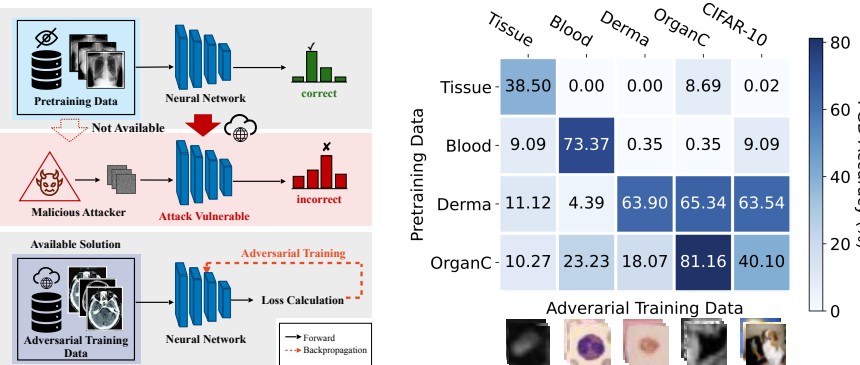

Figure 1: Motivational experiment using MedMNISTv2. Lefthand side demonstrates the problem scenario where adversarial threat prevails for models pretrained with private datasets. With no known solution at hand, using a similar dataset as an alternative for extra adversarial training is the only possible option. The righthand side plots the results when adversarial training is done with a similar or public dataset, which is shown to be ineffective in most cases.

where $\mathcal{L}_{class}$ is the classic cross-entropy loss, $\mathcal{L}_{feature}$ regularizes the samples' distributions ($\mu$, $\sigma$) to follow the saved statistics in the batch normalization layer ($\mu^T$, $\sigma^T$), and $\mathcal{L}_{prior}$ penalizes the total variance of the samples in the pixel level. These loss terms are jointly used to train a generator (Xu et al., 2020; Liu et al., 2021b; Choi et al., 2021; 2022) or to directly optimize samples from noise (Wang et al., 2021; Yin et al., 2020; Ghiasi et al., 2022), with fixed coefficients $\alpha_i$:

$$\mathcal{L}_{Synth} = \alpha_1 \mathcal{L}_{class} + \alpha_2 \mathcal{L}_{feature} + \alpha_3 \mathcal{L}_{prior}. \tag{6}$$

## 3 ADVERSARIAL ROBUSTNESS WITHOUT TRAINING DATA

### 3.1 PROBLEM DEFINITION

In the problem of learning data-free adversarial robustness, the objective of Equation (1) cannot be directly applied because none of $x$ or $y$ is available for training or fine-tuning. Instead, we are given an original model $T(\cdot)$ pretrained with $(x, y)$ without adversarial robustness, and the goal is to learn a robust model $S(\cdot)$. Hereafter, we will denote $T(\cdot)$ and $S(\cdot)$ as teacher and student, respectively.

As a common choice of data-free learning, we choose to use a surrogate training dataset $(\hat{x}, \hat{y})$ to train $S(\cdot)$, which allows us to use the de facto standard method for adversarial robustness: adversarial training. With the given notations we can reformulate the objective in Equation (1) as:

$$\min_\theta \frac{1}{n} \sum_i^n \max_{\hat{x}'_i \in \hat{\mathcal{X}}} \mathcal{L}(S_\theta(\hat{x}'_i), \hat{y}_i), \text{ where } \hat{\mathcal{X}} = \{\hat{x}'_i | \, \|\hat{x}'_i - \hat{x}_i\|_p \le \epsilon\}. \tag{7}$$

However, it remains to be answered how to create good surrogate training samples $(\hat{x}, \hat{y})$, and what loss function $\mathcal{L}$ can best generalize the learned robustness to defend against attacks on real data.

### 3.2 MOTIVATIONAL STUDY

Here, we demonstrate the difficulty of the problem by answering one naturally arising question: **can we just use another real dataset?** A relevant prior art is DAD (Nayak et al., 2022) which uses an auxiliary model trained with Tiny-ImageNet (Le & Yang, 2015) to defend against CIFAR-10. However, they strongly rely on the fact that these datasets are from the same domain. In practice, there is no guarantee on the similarity, especially on tasks with specific domains (e.g., biomedical).

Figure 1 (left) denotes the overall design of the motivational experiment, using categorized biomedical image datasets from MedMNIST v2 collection (Yang et al., 2023). Assuming the absence of the original dataset used for pretraining a given model, we use another dataset in the collection for additional adversarial training steps (Madry et al., 2018). Due to the different label spaces, we use teacher outputs as soft labels (i.e., $KL(S(x') \| T(x))$).

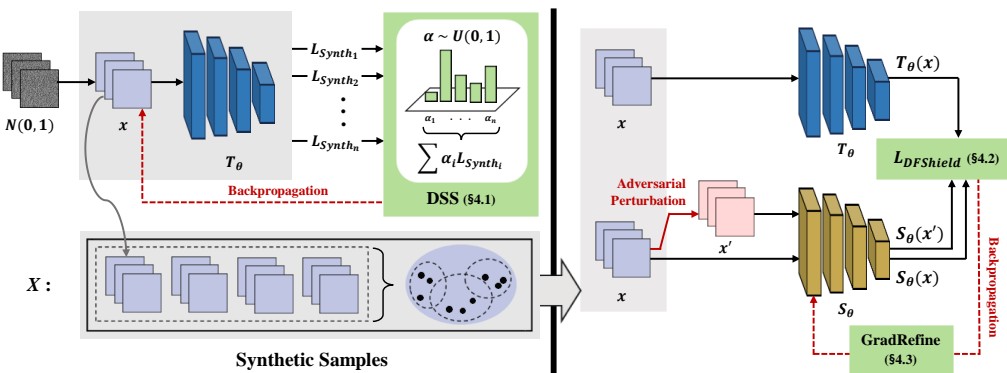

Figure 2: Procedure of the proposed method. Lefthand side denotes the proposed DSS. The righthand side shows adversarial training of target model $S_\theta$ using $\mathcal{L}_{DFShield}$ and GradRefine.

Figure 1 (right) shows the PGD-10 ($l_\infty$, $\epsilon = 8/255$) evaluation results using ResNet-18. Each row represents the dataset used for (non-robustly) pretraining a model, and each column represents the dataset used for additional adversarial training steps to gain robustness. It is clear that models adversarially trained using alternative datasets show poor robustness compared to those trained using the original dataset (the diagonal cells). Although there exist a few cases with high robustness from other datasets (e.g., OrganC → Derma), this is observed only between rare combinations. Moreover, using a relatively larger dataset (CIFAR-10) does not perform well either, which indicates that adversarial robustness is difficult to obtain from other datasets, even from larger ones.

## 4 DATAFREESHIELD: LEARNING DATA-FREE ADVERSARIAL ROBUSTNESS

To tackle the data-free adversarial robustness problem, we propose *DataFreeShield*, an effective solution to improve the robustness of the target model, illustrated in Figure 2. Overall, we generate a synthetic surrogate dataset (left) and use it to adversarially train $S_\theta$ initialized with $T_\theta$ (right). For generation, we propose diversified sample synthesis for dataset diversity (§4.1). For training, we propose a novel objective function (§4.2) and a gradient refinement (§4.3). We provide the pseudo-code of the algorithm in Appendix B.

### 4.1 DIVERSIFIED SAMPLE SYNTHESIS FOR MAXIMIZING DIVERSITY

The sample diversity is considered an extremely important factor to adversarial robustness (Sehwag et al., 2021; Rebuffi et al., 2021). Unfortunately, diversity is also known to be difficult to achieve with synthetic images, which can be exemplified by mode collapse phenomenon (Thanh-Tung & Tran, 2020; Srivastava et al., 2017; Mao et al., 2019) of generative models.

Instead of using generative models, we propose to use direct optimization with a novel diversifying technique called *diversified sample synthesis* (DSS). Direct optimization does not train a generative model, but directly updates each sample through backpropagation using an objective function ($\mathcal{L}_{Synth}$) (Yin et al., 2020; Cai et al., 2020; Zhong et al., 2022). To enhance the diversity of the samples, we dynamically modulate the synthesis loss $\mathcal{L}_{Synth}$. We first formulate $\mathcal{L}_{Synth}$ as a weighted sum of multiple loss terms. Then the weights are randomly set every iteration, letting each batch have a distinct distribution. Given a set $\mathbb{S} = \{\mathcal{L}_{Synth_1}, \mathcal{L}_{Synth_2}, ..., \mathcal{L}_{Synth_n}\}$, the conventional approach is to use their weighted sum with a fixed set of hyperparameters as in Equation (6). On the other hand, we use coefficients differently sampled for every batch from a continuous space:

$$\mathcal{L}_{Synth} = \sum_{i=1}^{|\mathbb{S}|} \alpha_i \mathcal{L}_{Synth_i}, \qquad \alpha_i \sim U(0, 1). \tag{8}$$

For the set $\mathbb{S}$, we use the three terms from Equation (6). The sampling of coefficients can follow any arbitrary distribution, where we choose a uniform distribution in this work.

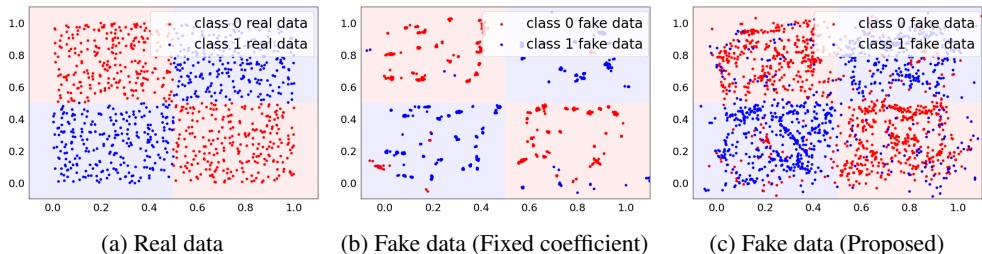

(a) Real data  (b) Fake data (Fixed coefficient)  (c) Fake data (Proposed)

Figure 3: Comparison of synthesis methods using the same number of 2-d data. The conventional fixed coefficient setting leads to limited diversity, while DSS generates diverse samples.

**A Toy Example.** To demonstrate the effectiveness DSS has on sample diversity, we conduct an empirical study on a toy example. Figure 3 displays the simplified experiment using 2-d data. The real data distribution is depicted in Figure 3a. Using the real data, we train a 4-layer neural network with batch normalization layers, which we use to synthesize fake data. Figure 3b demonstrates the results from conventional approaches (fixed coefficients following (Yin et al., 2020)). Although the data generally follows class information, they are highly clustered with small variance. On the other hand, Figure 3c shows the data generated using DSS, which are highly diverse and exhibit coverage much closer to that of the real data distribution.

## 4.2 TRAINING OBJECTIVE FUNCTION

Once a synthetic surrogate dataset has been generated, there exist several objective functions for adversarial training (Zhang et al., 2019; Wang et al., 2019; Goldblum et al., 2020; Zi et al., 2021). However, those objective functions mostly rely on the hard label $y$ of the dataset. Unfortunately, there is an inevitable dissimilarity in the synthetic samples compared to real ones, regardless of the quality. In such circumstances, relying on these artificial labels for adversarial training could convey incorrect guidance. To address this issue, we devise a new objective function $\mathcal{L}_{DFShield}$ that does not rely on the hard label, but only utilizes the soft guidance from the original model $T(\hat{x})$.

$$\mathcal{L}_{Train} = \mathcal{L}_{DFShield} = \underbrace{KL(S(\hat{x}), T(\hat{x}))}_{\text{(a) clean accuracy}} + \lambda_1 \underbrace{KL(S(\hat{x}'), T(\hat{x}))}_{\text{(b) adversarial robustness}} + \lambda_2 \underbrace{KL(S(\hat{x}'), S(\hat{x}))}_{\text{(c) smoothness}}. \quad (9)$$

The first term (a) optimizes the classification performance on clean samples, and can be thought as a replacement for the cross-entropy term from the common loss functions. The second term (b) serves the purpose of learning adversarial robustness similar to $\mathcal{L}_{CE}(S(\hat{x}'), y)$ used in standard AT (Equation (1)). The last term (c) trains the target model to be stable under small perturbations. Aside from not relying on artificial labels, using soft labels is also known to exhibit the benefit of leading the model to smoother minima (Choi et al., 2022; Yuan et al., 2020).

## 4.3 GRADIENT REFINEMENT FOR SMOOTHER LOSS SURFACE

With an obviously large gap between synthetic and real datasets, the minima reached using the synthetic data is unlikely to align well with that of the real data. In such a case, targeting a smoother loss surface is one promising approach. Inspired by a few techniques from federated learning and domain generalization (Tenison et al., 2022; Mansilla et al., 2021), we propose *GradRefine*, a novel gradient refinement method based on a parameter-wise agreement score. After computing gradients $g$ from $\mathcal{B}$ mini-batches, we calculate the agreement score $A_k$ for each parameter $k$ as:

$$A_k = \frac{1}{\mathcal{B}} \sum_{b=1}^{\mathcal{B}} sign(g_k^{(b)}). \quad (10)$$

Intuitively, $A$ denotes the amount which one sign outwins the other. $A$ is bounded by [-1,1], where $A = 0$ means equal distribution in both signs (maximum disagreement), and $A = \pm 1$ means one sign completely outwins the other (maximum agreement). Using $A$, we compute the final gradient $g_k^*$ that will be used for parameter $k$ update:

$$g_k^* = \Phi(A_k) \sum_{b=1}^{\mathcal{B}} \mathbb{1}_{\{A_k \cdot g_k^{(b)} > 0\}} \cdot g_k^{(b)}, \qquad \Phi(A_k) = \begin{cases} 1, & \text{if } |A_k| \geq \tau, \\ 0, & \text{otherwise}, \end{cases} \quad (11)$$

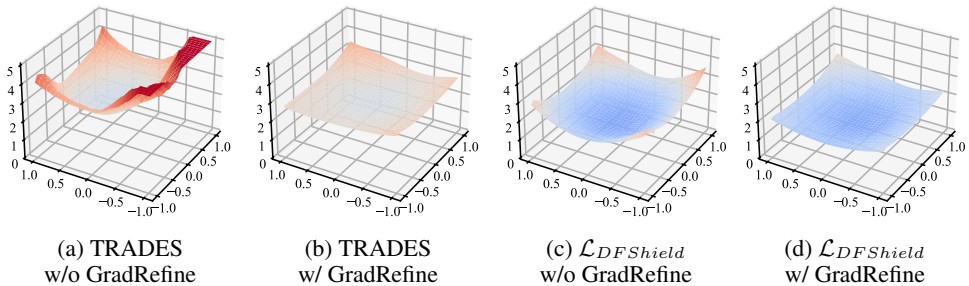

| (a) TRADES w/o GradRefine | (b) TRADES w/ GradRefine | (c) $\mathcal{L}_{DFShield}$ w/o GradRefine | (d) $\mathcal{L}_{DFShield}$ w/ GradRefine |

Figure 4: Loss surface visualization on ResNet-20 with CIFAR-10 showing that GradRefine achieves flatter loss surfaces. Each figure represents different training losses with or without GradRefine. We use normalized random direction for $x,y$ axis, following Li et al. (2018).

where $\mathbb{1}(\cdot)$ is the indicator function, and $\Phi$ is a masking function. We fix the threshold $\tau$ to be 0.5, which indicates that one should outwin the other for more than half its entirety. While high-fluctuating parameters are ignored by $\Phi$, we further pursue alignment by selectively using the agreeing gradient elements. Figure 4 visualizes the effect of GradRefine on the loss surface. In both TRADES and $\mathcal{L}_{DFShield}$, GradRefine yields a flatter loss surface, contributing towards better performance. Please refer to Appendix H for the visualization results of other cases.

## 5 EVALUATION

### 5.1 EXPERIMENTAL SETUP

We use total of four datasets, MedMNIST v2 (Yang et al., 2023), SVHN (Netzer et al., 2011), CIFAR-10, and CIFAR-100 (Krizhevsky et al., 2009) for evaluation. For MedMNIST v2, we use ResNet-18 and ResNet-50 originally trained by the authors and $l_\infty, \epsilon = 8/255$ perturbation budget. For CIFAR and SVHN datasets, we chose three pretrained models from PyTorchCV (pyt) library: ResNet-20, ResNet-56 (He et al., 2016), and WRN28-10 (Zagoruyko & Komodakis, 2016). We use $l_\infty, \epsilon = 4/255$ perturbation budget for SVHN, CIFAR-10, and CIFAR100, and additionally examine $l_2, \epsilon = 128/255$ setting. For results on extensive perturbation settings, please refer to Appendix D. For evaluation, AutoAttack (Croce & Hein, 2020) accuracy (denoted $\mathcal{A}_{\mathbf{AA}}$) is generally perceived as the standard metric (Croce et al., 2020). While we regard $\mathcal{A}_{\mathbf{AA}}$ as the primary interest, we also report the clean accuracy ($\mathcal{A}_{Clean}$) and PGD-10 accuracy ($\mathcal{A}_{\mathbf{PGD}}$) for interested readers. Further details of experimental settings can be found in Appendix A.

### 5.2 BASELINES

Since our work is the first to tackle the problem of data-free adversarial robustness, it is important to set an adequate baseline for comparison. We choose four of the most relevant works of other data-free learning tasks that generate synthetic samples to replace the original: DaST (Zhou et al., 2020) from a black-box at-

Table 1: Loss functions of baseline approaches

| Baselines | $\mathcal{L}_{Synth}$ | $\mathcal{L}_{train}$ |
|---|---|---|
| DaST | $-\mathcal{L}_{CE}(S(x), y)$ | $\mathcal{L}_{CE}(S(x'), y)$ |
| DFME | $-\sum\|T(x) - S(x)\|$ | $\sum\|T(x) - S(x')\|$ |
| AIT | $\mathcal{L}_{feature} + \mathcal{L}_{CE}(T(x), y)$ | $\mathcal{L}_{TRADES}$ |
| DFARD | $-\mathcal{L}_{KL}(S(x), T(x), \tilde{\tau})$ | $\mathcal{L}_{KL}(S(x'), T(x), \tilde{\tau})$ |

tack method, DFME (Truong et al., 2021) from data-free model extraction, AIT (Choi et al., 2022) from data-free quantization, and DFARD (Wang et al., 2023b) from data-free robust distillation. Since all four of these methods are not designed specifically for the data-free adversarial robustness problem, we adapt the training objective, summarized in Table 1. For details, please refer to the experimental settings in Appendix A.4.

### 5.3 PERFORMANCE COMPARISON

**MedMNIST v2.** Table 2 shows experimental results for MedMNIST v2, compared against the baseline methods (§5.2). The experiments represent a scenario close to real life where classification

Table 2: Performance on MedMNISTv2 with $l_\infty$ perturbation budget

| Dataset | Data-free | Method | ResNet-18 | | | ResNet-50 | | |
|---|---|---|---|---|---|---|---|---|
| | | | $\mathcal{A}_{Clean}$ | $\mathcal{A}_{\textbf{PGD}}$ | $\mathcal{A}_{\textbf{AA}}$ | $\mathcal{A}_{Clean}$ | $\mathcal{A}_{\textbf{PGD}}$ | $\mathcal{A}_{\textbf{AA}}$ |
| Tissue | ✗ | Public (CIFAR-10) | 22.04 | 0.02 | 0.00 | 27.84 | 10.11 | 8.64 |
| | ✓ | DaST | 9.95 | 1.20 | 0.68 | 27.15 | 0.35 | 0.02 |
| | | DFME | 23.69 | 23.40 | 0.03 | 7.13 | 0.45 | 0.19 |
| | | AIT | 28.48 | 0.38 | 0.00 | 22.65 | 0.11 | 0.01 |
| | | DFARD | 23.21 | 1.13 | 0.00 | 8.10 | 0.54 | 0.11 |
| | | **DataFreeShield** | 32.07 | **31.93** | **31.83** | 47.88 | **23.65** | **21.18** |
| Blood | ✗ | Public (CIFAR-10) | 9.09 | 9.09 | 0.00 | 9.09 | 9.09 | 0.00 |
| | ✓ | DaST | 9.12 | 8.65 | 7.51 | 90.64 | 1.11 | 0.06 |
| | | DFME | 91.23 | 1.37 | 0.03 | 93.95 | 0.32 | 0.00 |
| | | AIT | 19.47 | 17.04 | 12.98 | 19.47 | 4.03 | 0.03 |
| | | DFARD | 7.54 | 3.12 | 1.12 | 8.14 | 5.17 | 0.01 |
| | | **DataFreeShield** | 49.34 | **19.24** | **18.77** | 53.14 | **24.17** | **20.11** |
| Derma | ✗ | Public (CIFAR-10) | 66.88 | 63.54 | 62.11 | 67.89 | 62.48 | 60.11 |
| | ✓ | DaST | 67.48 | 50.02 | 42.39 | 67.08 | 44.04 | 34.11 |
| | | DFME | 11.12 | 11.12 | 11.12 | 66.88 | **66.88** | 63.12 |
| | | AIT | 45.23 | 5.04 | 4.69 | 59.75 | 16.61 | 13.67 |
| | | DFARD | 24.97 | 6.83 | 4.13 | 25.70 | 16.90 | 12.45 |
| | | **DataFreeShield** | 66.98 | **66.83** | **66.63** | 67.03 | 65.03 | **64.66** |
| OrganC | ✗ | Public (CIFAR-10) | 79.41 | 40.10 | 36.53 | 84.41 | 46.12 | 43.44 |
| | ✓ | DaST | 75.90 | 29.90 | 28.56 | 80.35 | 30.00 | 29.00 |
| | | DFME | 88.59 | 40.01 | 38.29 | 76.15 | 26.50 | 24.98 |
| | | AIT | 28.10 | 10.85 | 8.24 | 43.08 | 7.83 | 5.32 |
| | | DFARD | 70.12 | 9.45 | 7.83 | 77.12 | 13.04 | 10.12 |
| | | **DataFreeShield** | 76.89 | **46.92** | **45.18** | 82.82 | **53.45** | **51.11** |

Table 3: Performance on SVHN, CIFAR-10, and CIFAR-100 with $l_\infty$ perturbation budget

| Dataset | Method | ResNet-20 | | | ResNet-56 | | | WRN28-10 | | |
|---|---|---|---|---|---|---|---|---|---|---|
| | | $\mathcal{A}_{Clean}$ | $\mathcal{A}_{\textbf{PGD}}$ | $\mathcal{A}_{\textbf{AA}}$ | $\mathcal{A}_{Clean}$ | $\mathcal{A}_{\textbf{PGD}}$ | $\mathcal{A}_{\textbf{AA}}$ | $\mathcal{A}_{Clean}$ | $\mathcal{A}_{\textbf{PGD}}$ | $\mathcal{A}_{\textbf{AA}}$ |
| SVHN | DaST | 20.66 | 13.90 | 7.06 | 20.20 | 19.59 | 19.65 | 20.15 | 19.17 | 14.57 |
| | DFME | 11.32 | 2.59 | 0.84 | 20.20 | 19.22 | 4.27 | 6.94 | 5.31 | 0.28 |
| | AIT | 91.45 | 37.87 | 24.74 | 86.65 | 45.45 | 38.96 | 83.89 | 40.45 | 33.06 |
| | DFARD | 20.11 | 15.94 | 19.68 | 19.58 | 15.43 | 0.00 | 92.32 | 13.08 | 0.01 |
| | **DataFreeShield** | 91.83 | **54.82** | **47.55** | 88.66 | **62.05** | **57.54** | 94.14 | **69.60** | **62.66** |
| CIFAR-10 | DaST | 10.00$^\dagger$ | 9.89 | 8.62 | 12.06 | 7.68 | 5.32 | 10.00$^\dagger$ | 9.65 | 2.85 |
| | DFME | 14.36 | 5.23 | 0.08 | 13.81 | 3.92 | 0.03 | 10.00$^\dagger$ | 9.98 | 0.05 |
| | AIT | 32.89 | 11.93 | 10.67 | 38.47 | 12.29 | 11.36 | 34.92 | 10.90 | 9.47 |
| | DFARD | 12.28 | 5.33 | 0.00 | 10.84 | 8.93 | 0.00 | 9.82 | 12.01 | 0.02 |
| | **DataFreeShield** | 74.79 | **29.29** | **22.65** | 81.30 | **35.55** | **30.51** | 86.74 | **51.13** | **43.73** |
| CIFAR-100 | DaST | 1.01$^\dagger$ | 0.99 | 0.95 | 1.13 | 0.72 | 0.34 | 1.39 | 0.66 | 0.18 |
| | DFME | 1.86 | 0.53 | 0.24 | 24.16 | 0.98 | 0.25 | 66.30 | 0.67 | 0.00 |
| | AIT | 7.92 | 2.51 | 1.39 | 9.68 | 2.97 | 2.04 | 22.21 | 3.11 | 1.28 |
| | DFARD | 66.59 | 0.02 | 0.00 | 69.20 | 0.26 | 0.00 | 82.03 | 1.10 | 0.00 |
| | **DataFreeShield** | 41.67 | **10.41** | **5.97** | 39.29 | **13.23** | **9.49** | 61.35 | **23.22** | **16.44** |

$^\dagger$Did not converge

models are used for specific domains in the absence of public datasets from the same/similar domains. In all cases, DataFreeShield achieves the best results under $\mathcal{A}_{AA}$ evaluation. One interesting observation is that the baselines often perform worse than simply using real-world datasets of different domains. In Derma and OrganC, using CIFAR-10 leads to some meaningful robustness. We posit that this is because those datasets share similar features with CIFAR-10. Nonetheless, DataFreeShield performs significantly better than models trained with CIFAR-10 in all cases.

**Larger Datasets.** In Table 3, the performance of DataFreeShield is compared against the baselines on SVHN, CIFAR-10, and CIFAR-100 datasets. DataFreeShield outperforms the baselines by a huge margin in all cases. The improvements reach up to tens of %p in $\mathcal{A}_{AA}$, revealing the effectiveness of DataFreeShield and that the result is not from gradient obfuscation (Croce & Hein, 2020). Aligned with previous findings (Schmidt et al., 2018; Huang et al., 2022), larger models (ResNet-20 → ResNet-56 → WRN28-10) tend to have significantly better robust accuracy of up to 21.08%p difference between ResNet-20 and WRN28-10 under AutoAttack. However, the baselines were often unable to take advantage of the large model capacity (e.g., 19.65% → 14.57% in ResNet-56 → WRN28-10 with DaST on SVHN), and we believe this is due to the limited diversity of their synthetic samples. A similar trend can be found from the experiments done with $l_2$ perturbation budgets as shown in Table 4, where we compare with AIT, the best-performing baseline from Table 3.

Table 4: Performance on SVHN, CIFAR-10, and CIFAR-100 with $l_2$ perturbation budget

| Dataset | Method | ResNet-20 | | | ResNet-56 | | | WRN28-10 | | |
|---|---|---|---|---|---|---|---|---|---|---|
| | | $\mathcal{A}_{Clean}$ | $\mathcal{A}_{\mathbf{PGD}}$ | $\mathcal{A}_{\mathbf{AA}}$ | $\mathcal{A}_{Clean}$ | $\mathcal{A}_{\mathbf{PGD}}$ | $\mathcal{A}_{\mathbf{AA}}$ | $\mathcal{A}_{Clean}$ | $\mathcal{A}_{\mathbf{PGD}}$ | $\mathcal{A}_{\mathbf{AA}}$ |
| SVHN | AIT | 92.34 | 40.19 | 26.63 | 86.83 | 36.44 | 28.31 | 82.56 | 20.17 | 11.59 |
| | **DataFreeShield** | 92.15 | **51.86** | **42.67** | 89.06 | **58.98** | **53.45** | 94.20 | **66.28** | **56.94** |
| CIFAR-10 | AIT | 24.49 | 7.85 | 2.68 | 47.98 | 12.69 | 0.49 | 57.85 | 13.78 | 10.66 |
| | **DataFreeShield** | 74.27 | **31.68** | **25.46** | 83.33 | **38.15** | **32.34** | 88.54 | **50.53** | **42.09** |
| CIFAR-100 | AIT | 35.63 | 0.33 | 0.01 | 42.89 | 1.05 | 0.19 | 31.84 | 0.79 | 0.00 |
| | **DataFreeShield** | 43.57 | **12.11** | **7.60** | 43.28 | **15.42** | **11.32** | 64.34 | **24.92** | **17.14** |

Table 5: Comparison of dataset diversification methods

| Category | Diversification Method | CIFAR-10 Accuracy | | | Diversity Metric | | | |
|---|---|---|---|---|---|---|---|---|
| | | $\mathcal{A}_{Clean}$ | $\mathcal{A}_{\mathbf{PGD}}$ | $\mathcal{A}_{\mathbf{AA}}$ | Recall ↑ | Coverage ↑ | NDB ↓ | JSD ↓ |
| Data-free Diversification | Qimera | 76.88 | 18.90 | 10.68 | 0.000 | 0.002 | 99 | 0.514 |
| | RDSKD | 10.00 | 10.00 | 10.00 | 0.000 | 0.001 | 98 | 0.658 |
| | IntraQ | 13.77 | 36.13 | 12.46 | 0.308 | 0.087 | **88** | 0.275 |
| Augmentation | None | 91.46 | 43.66 | 36.34 | 0.535 | 0.101 | 91 | 0.253 |
| | Mixup | 90.61 | 48.16 | 36.43 | 0.641 | 0.084 | 94 | 0.322 |
| | Cutout | 92.59 | 39.84 | 34.39 | 0.535 | 0.034 | 95 | 0.443 |
| | CutMix | 91.90 | 42.79 | 34.79 | **0.845** | 0.084 | 93 | 0.328 |
| **Proposed** | **DSS** | 88.16 | **50.13** | **41.40** | 0.830 | **0.163** | **88** | **0.211** |

## 5.4 In-depth Study on DataFreeShield

**Training Loss.** Table 6 compares our proposed train loss against state-of-the-art ones used in adversarial training. STD (Madry et al., 2018), TRADES (Zhang et al., 2019), and MART (Wang et al., 2019) are from general adversarial training literature, while ARD (Goldblum et al., 2020) and RSLAD (Zi et al., 2021) are from robust distillation methods. Interestingly, MART provides almost no robustness in our problem. MART encourages learning from misclassified samples, which may lead the

Table 6: Comparison of $\mathcal{L}_{Train}$ on WRN-28-10

| $\mathcal{L}_{Train}$ | SVHN | | | CIFAR-10 | | |
|---|---|---|---|---|---|---|
| | $\mathcal{A}_{Clean}$ | $\mathcal{A}_{\mathbf{PGD}}$ | $\mathcal{A}_{\mathbf{AA}}$ | $\mathcal{A}_{Clean}$ | $\mathcal{A}_{\mathbf{PGD}}$ | $\mathcal{A}_{\mathbf{AA}}$ |
| AT | 93.71 | 69.32 | 62.58 | 81.63 | 48.03 | 38.94 |
| TRADES | 94.12 | 69.10 | 61.75 | 79.61 | 45.86 | 37.08 |
| MART | 35.94 | 2.55 | 1.09 | 13.69 | 6.74 | 0.09 |
| ARD | 96.29 | 61.11 | 52.56 | 90.95 | 36.61 | 31.16 |
| RSLAD | 96.03 | 64.59 | 57.04 | 90.25 | 39.30 | 31.16 |
| $\mathcal{L}_{\mathbf{DFShield}}$ | 94.87 | **69.67** | **65.66** | 88.16 | **50.13** | **41.40** |

model to overfit on low-quality synthetic samples. On the other hand, $\mathcal{L}_{DFShield}$ achieves the best results under both PGD-10 and AutoAttack in both datasets. The trend is consistent across different datasets and models, which we include in Appendix E.

**Dataset Diversification.** Table 5 compares DSS with other existing methods for dataset diversification. On one hand, we choose three data-free synthesis baselines for comparison: Qimera (Choi et al., 2021), IntraQ (Zhong et al., 2022), and RDSKD (Han et al., 2021). We additionally test three image augmentation methods, Mixup (Huang et al., 2020), Cutout (DeVries & Taylor, 2017), and CutMix (Yun et al., 2019) on top of direct sample optimization (Yin et al., 2020). It is clear that DSS outperforms all other diversification methods in terms of $\mathcal{A}_{AA}$.

For further investigation, we measure several well-known diversity metrics often used in evaluating generative models: recall, coverage (Naeem et al., 2020), number of statistically-different bins (NDB) (Richardson & Weiss, 2018), and Jensen-Shannon divergence (JSD). In almost all metrics, DSS shows the highest diversity, explaining its performance benefits. Although CutMix (Yun et al., 2019) shows slightly better recall than DSS, the difference is negligible and the coverage metric is generally perceived as a more exact measure of distributional diversity (Naeem et al., 2020). Measures on other datasets and models are included in Appendix E.

**Ablation Study.** Table 7 shows an ablation study of DataFreeShield. The baseline where none of our methods are applied denotes using the exact same set of synthesis loss functions without DSS, and adversarial training is done via TRADES. Across all models, there is a consistent gain under both PGD and especially more on AutoAttack. When applying $\mathcal{L}_{DFShield}$, there are cases where there is a minor drop in PGD accuracy but high gain on AutoAttack accuracy. This is due to $\mathcal{L}_{DFShield}$

Table 7: Ablation Study of DataFreeShield on CIFAR-10 dataset

| Model | $\mathcal{L}_{DFShield}$ | DSS | GradRefine | $\mathcal{A}_{Clean}$ | $\mathcal{A}_{PGD}$ | $\mathcal{A}_{AA}$ |
|---|---|---|---|---|---|---|
| ResNet-20 | ✗ | ✗ | ✗ | 86.42 | 26.73 | 2.03 |
| | ✓ | ✗ | ✗ | 82.58 | 23.93 (-2.80) | 14.61 (+12.58) |
| | ✓ | ✓ | ✗ | 77.83 | 27.42 (+0.69) | 19.09 (+17.06) |
| | ✓ | ✓ | ✓ | 74.79 | **29.29 (+2.56)** | **22.65 (+20.62)** |
| ResNet-56 | ✗ | ✗ | ✗ | 78.22 | 34.44 | 24.34 |
| | ✓ | ✗ | ✗ | 83.72 | 30.91 (-3.53) | 27.42 (+3.08) |
| | ✓ | ✓ | ✗ | 83.67 | 34.78 (+0.34) | 27.69 (+3.35) |
| | ✓ | ✓ | ✓ | 81.30 | **35.55 (+1.11)** | **30.51 (+6.17)** |
| WRN28-10 | ✗ | ✗ | ✗ | 80.29 | 42.51 | 37.96 |
| | ✓ | ✗ | ✗ | 91.46 | 43.66 (+1.15) | 36.34 (-1.62) |
| | ✓ | ✓ | ✗ | 88.16 | 50.13 (+7.62) | 41.40 (+3.44) |
| | ✓ | ✓ | ✓ | 86.74 | **51.13 (+8.62)** | **43.73 (+5.77)** |

effectively reducing the gap between relatively weaker and stronger attacks. GradRefine adds a similar improvement, resulting in 6.17%p to 20.62%p gain altogether under AutoAttack.

## 6 RELATED WORK

**Adversarial Defense.** Existing defense methods train robust classifiers by feeding perturbed data to the model. Popular approaches include specially designing loss functions as variants of AT (Madry et al., 2018), such as TRADES (Zhang et al., 2019) or MART (Wang et al., 2019). A recent trend is to import extra data from other datasets (Carmon et al., 2019; Rebuffi et al., 2021), or generated under the supervision of real data (Rebuffi et al., 2021; Sehwag et al., 2021). However, such rich datasets are not easy to obtain in practice, sometimes none is available as in the problem we target.

**Data-free Learning.** Training or fine-tuning an existing model in absence of data has been studied to some degree. However, most are related to, or confined to only compression tasks, some of which are knowledge distillation (Fang et al., 2019; Lopes et al., 2017), pruning (Srinivas & Babu, 2015), and quantization (Nagel et al., 2019; Cai et al., 2020; Xu et al., 2020; Liu et al., 2021b; Choi et al., 2021; 2022; Zhu et al., 2021). A concurrent work DFARD (Wang et al., 2023b) sets a similar but different problem where a robust model already exists, and the objective is to distill it to a lighter network. Without the existence of a robust model, the effectiveness of DFARD is significantly reduced.

**Gradient Refining Techniques.** Adjusting gradients is an effective technique often used for diverse purposes. Yu et al. (2020) directly projects gradients with opposing directionality to dominant task gradients before model update. Liu et al. (2021a) selectively uses gradients that can best aid the worst performing task, and Fernando et al. (2022) estimates unbiased approximations of gradients to ensure convergence in various initializations. Eshratifar et al. (2018) also utilizes gradients to maximize generalization ability to unseen data under a certain task. Similar to ours, Mansilla et al. (2021) and Tenison et al. (2022) updates the model based on sign agreement of gradients across domains or clients. Shi et al. (2022), Wang et al. (2023a), and Dandi et al. (2022) maximize gradient inner product between different domains or loss terms to promote gradient alignment.

## 7 CONCLUSION

For the first time, we define the problem of learning data-free adversarial robustness, and propose DataFreeShield, an effective method for instilling robustness to a given model using synthetic data. We approach the problem from two perspectives of generating diverse synthetic datasets and training with flatter loss surfaces. Experimental results show that DataFreeShield significantly outperforms baseline approaches, demonstrating that robustness can be achieved without the original datasets.

**Limitation and Future Work.** The performance gap between data-free methods and data-driven have always been an agonizing pain in most data-free literature (Nagel et al., 2020; 2019; Cai et al., 2020; Xu et al., 2020; Nayak et al., 2019). Although our method achieves noticeable gains over the baselines, there still exists large room for improvements, especially with datasets of higher complexity (larger resolution, number of classes, *etc.*).

ETHICS STATEMENT

As shown in Appendix J, the generated samples are not very human-recognizable, and being so does not necessarily lead to better performance of the models. From these facts, we believe our synthetic input generation does not cause privacy invasion that might have existed from the original training dataset. However, there is still a possibility where the generated samples could affect privacy concerns, such as membership inference attacks (Shokri et al., 2017) or model stealing (Lee et al., 2019). For example, an attacker might compare the image-level or feature-level similarity of some test samples with the synthetically generated samples to find out whether the test sample is part of the training set or not. We believe further investigation is needed on such side-effects, which we leave as a future work.

REPRODUCIBILITY STATEMENT

To ensure reproducibility, we have described the experimental details in Section 5.1 and Appendix A. All the code used for the experiment has been submitted in the supplementary material as a zip archive, along with the scripts for reproduction. If there are further questions or issues regarding reproduction in any of the presented result in the future, we will faithfully address them.

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

APPENDIX

We provide a more extensive set of experimental results with some analyses that we could not include in the main body due to space constraints. The contents of this material are as below:

- **Detailed Experimental Settings (Appendix A)**: We provide detailed information of our experiments.
- **Overall Procedure of DataFreeShield (Appendix B)**: We provide pseudo-code of the overall procedure of DataFreeShield.
- **Number of Synthetic Samples (Appendix C)**: We study the effect of using different numbers of synthetic samples.
- **Extended Set of Experiments on $\epsilon$-bounds (Appendix D)**: Extended results on diverse attack distance are presented.
- **Detailed Study on DataFreeShield (Appendix E)**: Extended results of detailed study of DataFreeShield on sample diversity and comparison against different training loss functions.
- **Comparison against Test-Time Defense Methods (Appendix F**: We compare existing test-time defense methods against DataFreeShield.
- **Evaluation under Adaptive Adversarial Attacks (Appendix G**: We evaluate robustness of DataFreeShield under adaptive attacks.
- **Further Visualization of Loss Surface (Appendix H)**: We provide further analysis on $\mathcal{L}_{DFShield}$ and its effect on the loss surface.
- **Sensitivity on the Number of Aggregated Batches (Appendix I)**: Selected examples of synthetic data are presented.
- **Generated Synthetic Data (Appendix J)**: Selected examples of synthetic data are presented.

## A   DETAILED EXPERIMENTAL SETTINGS

In this section, we provide details on experimental settings for both synthetic data generation and robust training. For baseline implementation of DaST (Zhou et al., 2020), DFME (Truong et al., 2021), and AIT (Choi et al., 2022), we used the original code from the authors, except for the modifications we specified in Appendix A.4. For DFARD (Wang et al., 2023b) we followed the description in the publication since the original implementation is not available, and used ACGAN (Odena et al., 2017) due to missing details of generator architecture in the original publication. All experiments have been conducted using PyTorch 1.9.1 and Python 3.8.0 running on Ubuntu 20.04.3 LTS with CUDA version 11.1 using RTX3090 and A6000 GPUs.

### A.1   CODE

The code used for the experiment is included in a zip archive in the supplementary material, along with the script for reproduction. The code is under Nvidia Source Code License-NC and GNU General Public License v3.0.

### A.2   DATA GENERATION

When optimizing gaussian noise, we use Adam optimizer with learning rate $= 0.1$ with batch size of 200, which we optimize for 1000 iterations. For diversified sample synthesis, we set the range [0,1] for sampling distribution of coefficients, and use uniform distribution. Code implementation for diversified sample synthesis builds upon a prior work (Yin et al., 2020). For MedMNIST v2 results, we generated 10,000 samples for training, and for the other datasets we used 60,000 samples. To accelerate data generation, we use multiple GPUs in parallel where 10,000 samples are generated with each. With batch size of 200, generating 10,000 samples of size 28x28 using ResNet-18 takes 0.7 hours on RTX 3090. For 32x32 sized samples, ResNet-20 takes 0.6 hours, ResNet-56 2.6 hours, and WRN28-10 3.6 hours.

A.3 Adversarial Training

For adversarial training, we used SGD optimizer with learning rate=1e-4, momentum=0.9, and batch size of 200 for 100 epochs, and 200 epochs for ResNet-20 and ResNet-18. All adversarial perturbations were created using PGD-10 (Madry et al., 2018) with the specified $\epsilon$-bounds. Following the convention, $l_2$-norm attacks are bounded by $\epsilon = 128/255$ with step size of $15/255$. $l_\infty$-norm attacks are evaluated under a diverse set of distances $\epsilon = \{8/255, 6/255, 4/255, 2/255\}$, which all use step size= $\epsilon/4$. For $\mathcal{L}_{DFShield}$, we simply use $\lambda_1 = 1$ and $\lambda_2 = 1$, which we found to best balance the learning from three different objective terms. For GradRefine, we use $\mathcal{B} = \{10, 20\}$ for all settings, which we found to perform generally well across different datasets and models. When using GradRefine, we increment learning rate linearly with $\mathcal{B}$ to take into consideration the increased effective batchsize. We use $\tau = 0.5$ for all our experiments with GradRefine.

A.4 Adaptation of the Baselines

In this section, we describe how we adapted the baselines (Table 1) to the problem of data-free adversarial robustness. DaST (Zhou et al., 2020) is a black-box attack method with no access to the original data. DaST trains a substitute model using samples from a generative model (Goodfellow et al., 2014) to synthesize samples for querying the victim model. To adapt DaST to our problem, we keep the overall framework but modify the training loss, substituting clean samples with perturbed ones. This makes it possible to use the training algorithm, while the objective now is to robustly train a model with no data.

DFME (Truong et al., 2021) is a more recent work on data-free model extraction that also utilizes synthetic samples for model stealing. They leverage distillation methods (Fang et al., 2019) to synthesize samples that maximize student-teacher disagreement. Similar to DaST, we substitute the student input to perturbed ones, while keeping other settings the same.

AIT (Choi et al., 2022) utilizes full precision model's feedback for training its generative model. Unlike DaST and DFME that focus on student outputs when training the generator, AIT additionally utilizes the batch-normalization statistics stored in the teacher model for creating synthetic samples. Since AIT is a model quantization method, its student model is of low-bit precision, and thus their training loss cannot be directly adopted to our task. We use TRADES (Zhang et al., 2019) loss function for training, a variation of AT (Madry et al., 2018).

Lastly, DFARD (Wang et al., 2023b) suggests data-free robust distillation. Given a model already robustly trained, the goal is to distill its robustness to a lighter network. They use adaptive distillation temperature to regulate learning difficulty. While this seems to align with the data-free adversarial robustness, the robust teacher is not available in our problem. Therefore, we replace the robustly pretrained model with the given (non-robust) $T(x)$ so that student can correctly classify perturbed samples.

B Overall Procedure of DataFreeShield

The pseudo-code of the overall procedure of DataFreeShield is depicted in Algorithm 1. It comprises data generation using *diversified sample synthesis* (line 4-10, §4.1), and adversarial training using a novel loss function (line 15, §4.2) along with a gradient refinement technique (line 17-20, §4.3).

C Number of Synthetic Samples

In this section, we show the performance gain from simply incrementing the number of synthetic samples. Figure 5 plots the AutoAttack accuracy when trained using differing number of samples. For all models, the trend is similar in that the performance increases linearly, and converge at some point around 50000-60000. Although there exists marginal gain with further supplement of data, we settle for 60000 samples for the training efficiency. One observation is that for smaller model (ResNet-20), it is much harder to obtain meaningful robustness for any set under 20000. We posit this is due to the characteristic of data-free synthesis, where the only guidance is from a pretrained model and the quality of the data is bounded by the performance of the pretrained model. Since larger models tend to learn better representation, it can be reasoned that the smaller models are less capable

---

**Algorithm 1** Procedure of DataFreeShield

---

1: **Inputs:** set of synthesis loss terms $\mathbb{S}$, number of batches for synthesis $N$, pretrained model's parameters $\theta_T$, target model for training $\theta_S$, synthesis iterations $Q$, train iterations $P$, number of aggregated batches $\mathcal{B}$, learning rate for synthesis $\eta_g$ and training $\eta_s$.
2: **Initialize:** $\theta_S \leftarrow \theta_T$         ▷ Initialize target model with pretrained model
3: **Initialize:** $X = \{X_1, ..., X_N\} \leftarrow Z \sim \mathcal{N}(0,1)$     ▷ Initialize batches with random noise
4: **for** $i$=1 ,..., $N$ **do**
5:   Sample $\{\alpha_1, ..., \alpha_{|\mathbb{S}|}\}$ from $\mathcal{U}(0,1)$
6:   $\mathcal{L}_{Synth} = \sum_{s=1}^{|\mathbb{S}|} \alpha_s * \mathcal{L}_s$       ▷ Diversified Sample Synthesis (§4.1)
7:   **for** $q$=1 ,..., $Q$ **do**
8:    $X_i \leftarrow X_i - \eta_g \nabla_{X_i} \mathcal{L}_{Synth}(X_i; \theta_T)$
9:   **end for**
10: **end for**
11: **for** $p = 1, ..., P$ **do**
12:   Sample $\mathcal{B}$ mini-batches $\{X_1, ..., X_{\mathcal{B}}\}$ from $X$
13:   **for** $b$=1 ,..., $\mathcal{B}$ **do**
14:    $X_b' \leftarrow PGD(X_b; \theta_S)$          ▷ Equation (2)
15:    $g^{(b)} \leftarrow \nabla_{\theta_S} \mathcal{L}_{DFShield}(X_b, X_b')$     ▷ $\mathcal{L}_{DFShield}$ (§4.2)
16:   **end for**
17:   **for** $k = 1, ..., |\theta_S|$ (in parallel) **do**
18:    $A_k = \frac{1}{\mathcal{B}} \sum_{b=1}^{\mathcal{B}} sign(g_k^{(b)})$       ▷ GradRefine (§4.3)
19:    $g_k^* = \Phi(A_k) \cdot \sum_{b=1}^{\mathcal{B}} \mathbb{1}_{\{A_k \cdot g_k^{(b)} > 0\}} \cdot g_k^{(b)}$     ▷ Equation (11)
20:   **end for**
21:   $\theta_S \leftarrow \theta_S - \eta_s g^*$         ▷ Update using refined gradient
22: **end for**

---

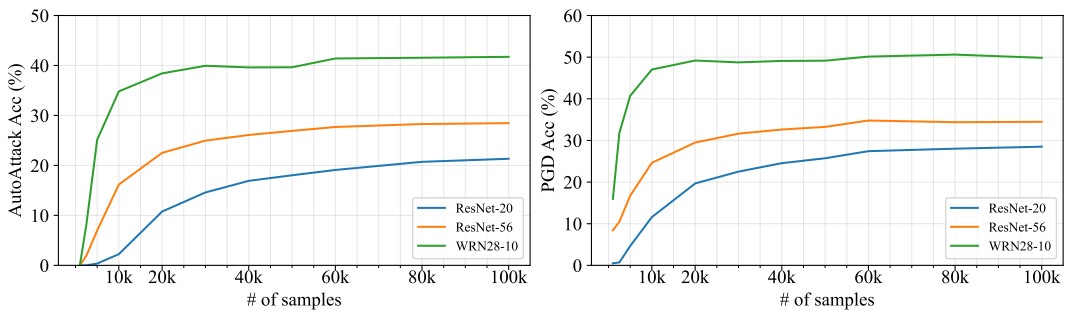

Figure 5: Comparing performance using varying number of samples for training. Left denotes AutoAttack accuracy while the right denotes PGD-10 accuracy.

of synthesizing good quality data, along with the reason that smaller models are generally harder to train for adversarial robustness.

## D  EXTENDED SET OF EXPERIMENTS ON $\epsilon$-BOUNDS

In the field of empirical adversarial robustness (Rebuffi et al., 2021; Schmidt et al., 2018; Wang et al., 2019; Wu et al., 2020), thorough evaluation under attacks of varying difficulties (number of iterations, size of $\epsilon$, etc) is needed to guarantee the model's robustness. This is because a consistent trend across different attacks and resistence against strong attacks (AutoAttack) ensures the robustness is not from obfuscated gradients (Athalye et al., 2018). In this regard, we provide further experiment results using diverse set of $\epsilon$-bounds using SVHN and CIFAR-10 in Table 8 and Table 9. For each setting, we highlight the best results under $\mathcal{A}_{AA}$.

In both datasets, baseline methods show poor performance regardless of the difficulty of the attack. For example, in CIFAR-10, even at a relatively weaker attack of $\epsilon = 2/255$, DaST, DFME, and DFARD do not exceed 10% under AutoAttack evaluation, which is no better than random guessing. Although AIT performs generally better than the other baselines, it suffers when training a larger

Table 8: Performance on SVHN

| $\epsilon$ | Method | ResNet-20 | | | ResNet-56 | | | WRN-28-10 | | |
|---|---|---|---|---|---|---|---|---|---|---|
| | | $\mathcal{A}_{Clean}$ | $\mathcal{A}_{\mathbf{PGD}}$ | $\mathcal{A}_{\mathbf{AA}}$ | $\mathcal{A}_{Clean}$ | $\mathcal{A}_{\mathbf{PGD}}$ | $\mathcal{A}_{\mathbf{AA}}$ | $\mathcal{A}_{Clean}$ | $\mathcal{A}_{\mathbf{PGD}}$ | $\mathcal{A}_{\mathbf{AA}}$ |
| | Original | 95.42 | 88.04 | 86.72 | 96.00 | 88.86 | 87.85 | 96.06 | 89.23 | 88.16 |
| 2/255 | DaST | 93.80 | 34.30 | 12.33 | 91.00 | 46.29 | 31.77 | 96.45 | 35.21 | 9.49 |
| | DFME | 96.05 | 35.24 | 8.39 | 97.30 | 38.79 | 10.98 | 97.21 | 24.67 | 0.54 |
| | AIT | 94.67 | 65.74 | 60.74 | 95.63 | 70.42 | 66.23 | 85.82 | 44.33 | 36.37 |
| | DFARD | 96.58 | 32.64 | 6.89 | 97.29 | 39.21 | 8.94 | 97.11 | 26.38 | 0.29 |
| | **DataFreeShield** | 94.22 | **75.56** | **72.17** | 94.16 | **80.32** | **78.47** | 95.94 | **84.63** | **82.93** |
| | Original | 93.19 | 78.01 | 74.59 | 94.67 | 79.53 | 79.67 | 94.48 | 79.53 | 76.72 |
| 4/255 | DaST | 20.66 | 13.90 | 7.06 | 20.20† | 19.59 | 19.65 | 20.15 | 19.17 | 14.57 |
| | DFME | 11.32† | 2.59 | 0.84 | 20.20† | 19.22 | 4.27 | 6.94† | 5.31 | 0.28 |
| | AIT | 91.45 | 37.87 | 24.74 | 86.65 | 45.45 | 38.96 | 83.89 | 40.45 | 33.06 |
| | DFARD | 20.11 | 15.94 | 19.68 | 19.58 | 15.43 | 0.00 | 92.32 | 13.08 | 0.01 |
| | **DataFreeShield** | 91.83 | **54.82** | **47.55** | 88.66 | **62.05** | **57.54** | 94.14 | **69.60** | **62.66** |
| | Original | 91.47 | 67.39 | 60.56 | 91.59 | 71.10 | 57.95 | 93.62 | 75.03 | 57.36 |
| 6/255 | DaST | 7.84 | 1.64 | 0.00 | 19.68 | 19.57 | 12.79 | 61.72 | 8.82 | 0.00 |
| | DFME | 15.90† | 15.94 | 14.81 | 97.34 | 5.21 | 0.00 | 97.11 | 1.39 | 0.00 |
| | AIT | 83.70 | 23.20 | 6.03 | 87.23 | 30.06 | 17.37 | 77.05 | 12.45 | 3.61 |
| | DFARD | 24.27 | 19.48 | 0.44 | 97.17 | 5.87 | 0.00 | 54.24 | 19.58 | 0.00 |
| | **DataFreeShield** | 89.00 | **39.63** | **31.15** | 81.90 | **47.36** | **40.88** | 92.18 | **55.39** | **45.57** |
| | Original | 86.50 | 55.68 | 40.31 | 89.29 | 59.39 | 51.21 | 92.03 | 68.35 | 32.94 |
| 8/255 | DaST | 10.29 | 3.94 | 2.07 | 19.68† | 19.59 | 19.68 | 20.39 | 16.69 | 1.35 |
| | DFME | 20.15 | 0.30 | 0.00 | 21.55 | 16.60 | 0.22 | 6.84† | 6.70 | 2.29 |
| | AIT | 47.47 | 15.21 | 7.70 | 73.33 | 22.42 | 10.92 | 47.96 | 14.85 | 7.24 |
| | DFARD | 20.03 | 13.46 | 0.00 | 25.18 | 5.46 | 0.00 | 93.07 | 18.23 | 0.02 |
| | **DataFreeShield** | 85.32 | **29.96** | **20.84** | 75.70 | **37.32** | **29.04** | 90.57 | **43.80** | **31.77** |

†Did not converge

Table 9: Performance on CIFAR-10

| $\epsilon$ | Method | ResNet-20 | | | ResNet-56 | | | WRN-28-10 | | |
|---|---|---|---|---|---|---|---|---|---|---|
| | | $\mathcal{A}_{Clean}$ | $\mathcal{A}_{\mathbf{PGD}}$ | $\mathcal{A}_{\mathbf{AA}}$ | $\mathcal{A}_{Clean}$ | $\mathcal{A}_{\mathbf{PGD}}$ | $\mathcal{A}_{\mathbf{AA}}$ | $\mathcal{A}_{Clean}$ | $\mathcal{A}_{\mathbf{PGD}}$ | $\mathcal{A}_{\mathbf{AA}}$ |
| | Original | 78.82 | 66.59 | 65.53 | 81.01 | 68.34 | 67.42 | 86.34 | 75.39 | 74.81 |
| 2/255 | DaST | 10.02† | 9.91 | 9.77 | 17.46 | 3.25 | 0.50 | 12.72 | 6.46 | 2.37 |
| | DFME | 32.05 | 9.60 | 4.19 | 91.29 | 16.28 | 0.13 | 97.32 | 18.31 | 0.00 |
| | AIT | 81.25 | 28.90 | 24.72 | 78.51 | 33.66 | 30.25 | 74.05 | 6.66 | 1.27 |
| | DFARD | 91.89 | 7.63 | 0.08 | 95.34 | 16.97 | 0.06 | 97.32 | 10.71 | 0.00 |
| | **DataFreeShield** | 80.66 | **50.09** | **46.73** | 87.06 | **57.99** | **55.44** | 91.56 | **70.48** | **68.12** |
| | Original | 75.60 | 56.79 | 54.37 | 76.76 | 58.50 | 56.54 | 82.89 | 64.64 | 62.84 |
| 4/255 | DaST | 10.00† | 9.89 | 8.62 | 12.06 | 7.68 | 5.32 | 10.00† | 9.65 | 2.85 |
| | DFME | 14.36 | 5.23 | 0.08 | 13.81 | 3.92 | 0.03 | 10.00† | 9.98 | 0.05 |
| | AIT | 32.89 | 11.93 | 10.67 | 38.47 | 12.29 | 11.36 | 34.92 | 10.90 | 9.47 |
| | DFARD | 12.28 | 5.33 | 0.00 | 10.84 | 8.93 | 0.00 | 9.82 | 12.01 | 0.02 |
| | **DataFreeShield** | 74.79 | **29.29** | **22.65** | 81.30 | **35.55** | **30.51** | 86.74 | **51.13** | **43.73** |
| | Original | 70.88 | 48.23 | 45.88 | 73.55 | 50.47 | 47.50 | 77.89 | 54.56 | 52.23 |
| 6/255 | DaST | 10.00† | 9.86 | 8.02 | 10.00† | 9.00 | 2.21 | 10.17 | 4.97 | 0.07 |
| | DFME | 10.00† | 0.82 | 0.01 | 78.82 | 3.35 | 0.00 | 10.86 | 9.26 | 1.58 |
| | AIT | 24.20 | 7.71 | 3.05 | 22.35 | 9.46 | 7.46 | 63.61 | 3.87 | 0.51 |
| | DFARD | 11.23 | 4.91 | 0.00 | 95.27 | 1.10 | 0.00 | 92.46 | 0.34 | 0.00 |
| | **DataFreeShield** | 69.11 | **17.94** | **11.03** | 76.55 | **21.55** | **16.11** | 81.26 | **37.26** | **26.07** |
| | Original | 69.19 | 41.69 | 37.30 | 70.79 | 43.89 | 39.97 | 76.76 | 47.88 | 44.04 |
| 8/255 | DaST | 10.00† | 9.99 | 6.81 | 10.60† | 9.18 | 1.62 | 10.00† | 9.88 | 0.56 |
| | DFME | 13.17 | 1.67 | 0.00 | 10.01† | 2.10 | 0.00 | 10.02† | 4.44 | 0.00 |
| | AIT | 14.02 | 3.49 | 0.28 | 10.06† | 9.97 | 9.96 | 10.12† | 9.66 | 8.16 |
| | DFARD | 11.23 | 1.41 | 0.00 | 13.04 | 3.41 | 0.00 | 10.11 | 9.98 | 0.00 |
| | **DataFreeShield** | 63.69 | **10.53** | **4.71** | 73.05 | **13.27** | **7.80** | 76.63 | **27.61** | **14.79** |

†Did not converge

Table 10: Comparison on SVHN

| Method | ResNet-20 | | | ResNet-56 | | | WRN-28-10 | | |
|---|---|---|---|---|---|---|---|---|---|
| | $\mathcal{A}_{Clean}$ | $\mathcal{A}_{\mathbf{PGD}}$ | $\mathcal{A}_{\mathbf{AA}}$ | $\mathcal{A}_{Clean}$ | $\mathcal{A}_{\mathbf{PGD}}$ | $\mathcal{A}_{\mathbf{AA}}$ | $\mathcal{A}_{Clean}$ | $\mathcal{A}_{\mathbf{PGD}}$ | $\mathcal{A}_{\mathbf{AA}}$ |
| AT | 23.34 | 16.73 | 13.83 | 95.12 | 42.66 | 8.73 | 93.71 | 69.32 | 62.58 |
| TRADES | 92.99 | 51.13 | 36.71 | 95.73 | **67.00** | 20.87 | 94.12 | 69.10 | 61.75 |
| MART | 63.36 | 6.48 | 1.98 | 91.65 | 26.09 | 4.74 | 35.94 | 2.55 | 1.09 |
| ARD | 94.78 | 43.10 | 30.38 | 96.02 | 47.37 | 37.16 | 96.29 | 61.11 | 52.56 |
| RSLAD | 93.75 | 44.06 | 29.81 | 94.25 | 56.60 | 48.40 | 96.03 | 64.59 | 57.04 |
| $\mathcal{L}_{\mathbf{DFShield}}$ (Proposed) | 91.78 | **54.53** | **45.50 (+8.79)** | 91.06 | 63.12 | **56.54 (+8.14)** | 94.87 | **69.67** | **65.66 (+3.08)** |

†Did not converge

Table 11: Comparison on CIFAR-10

| Method | ResNet-20 | | | ResNet-56 | | | WRN-28-10 | | |
|---|---|---|---|---|---|---|---|---|---|
| | $\mathcal{A}_{Clean}$ | $\mathcal{A}_{\mathbf{PGD}}$ | $\mathcal{A}_{\mathbf{AA}}$ | $\mathcal{A}_{Clean}$ | $\mathcal{A}_{\mathbf{PGD}}$ | $\mathcal{A}_{\mathbf{AA}}$ | $\mathcal{A}_{Clean}$ | $\mathcal{A}_{\mathbf{PGD}}$ | $\mathcal{A}_{\mathbf{AA}}$ |
| AT | 23.51 | 6.09 | 1.66 | 92.49 | **46.38** | 0.12 | 81.63 | 48.03 | 38.94 |
| TRADES | 86.34 | 26.81 | 1.75 | 81.71 | 29.49 | 9.36 | 79.61 | 45.86 | 37.08 |
| MART | 14.91 | 2.67 | 0.22 | 91.65 | 16.23 | 0.00 | 13.69 | 6.74 | 0.09 |
| ARD | 90.13 | 9.83 | 0.17 | 92.21 | 9.31 | 2.51 | 90.95 | 36.61 | 31.16 |
| RSLAD | 77.85 | 11.66 | 0.69 | 88.98 | 19.59 | 12.27 | 90.25 | 39.30 | 31.16 |
| $\mathcal{L}_{\mathbf{DFShield}}$ (Proposed) | 77.83 | **27.42** | **19.09 (+17.34)** | 83.67 | 34.78 | **27.69 (+15.42)** | 88.16 | **50.13** | **41.40 (+2.46)** |

†Did not converge

model (WRN-28-10). The overall trend of the baselines implies that these methods are unable to learn meaningful robustness, regardless of the size of the distortion. On the other hand, DataFreeShield shows consistent trend across all attacks. While exceeding the baseline methods by a huge margin, the results are stable under both PGD and AutoAttack in all $\epsilon$'s. This shows that DataFreeShield is able to learn meaningful robustness from adversarial training of all presented distortion sizes.

# E    DETAILED STUDY ON DATAFREESHIELD

We present extended version of detailed study presented in the main paper. Table 10 and Table 11 compare state-of-the-art AT loss functions against our proposed $\mathcal{L}_{DFShield}$. The results are consistent with what we have displayed in the main paper, where $\mathcal{L}_{DFShield}$ performs the best in almost all settings. Although the other loss functions perform generally well in WRN-28-10, they tend to fall into false sense of security with ResNet-20 and ResNet-56, where the seemingly robust models under weak attacks (PGD) easily break under stronger attacks (AutoAttack). For example, in ResNet-56 of Table 11, Standard AT (Madry et al., 2018) achieves 46.38% under PGD, but is easily circumvented by AutoAttack, which gives 0.12%. Similar phenomenon is observed across other loss functions. However, $\mathcal{L}_{DFShield}$ is consistent under both PGD and AutoAttack, and shows no sign of obfuscated gradients.

For comparison, we present real-data training performance on the MedMNISTv2 dataset in Table 12. The 'original' data training uses the exact same domain for adversarial training, so that can be regarded as the upper bound of the data-free adversarial robustness. The experimental results show that even real data from another domain (CIFAR-10) significantly underperform compared to the original dataset. On the other hand, DataFreeShield shows superior performance than the other-domain public dataset. Remarkably, DataFreeShield almost reached similar performance levels with the original dataset training in the Derma dataset. The experimental results show the advantages of DataFreeShield, by reducing the gap towards real-data training.

Similarly for dataset diversification, we show extended version in Table 13 and Table 14. In all settings, diversified sample synthesis shows best quantitative measure under Coverage and JSD. Coverage is known to be a more accurate measure of diversity than Recall in the sense that it is more robust against outliers (Naeem et al., 2020). Also, JSD measures distributional distance, which is frequently used in evaluating GANs. Thus, they show quantitative evidence to diversity gain of diversified sample synthesis. This aligns with the robust training results, where diversified sample synthesis outperforms other diversifying methods in most settings.

Table 12: Real-data training performance of MedMNISTv2 with $l_\infty$ perturbation budget

| Dataset | Data-free | Method | ResNet-18 | | | ResNet-50 | | |
|---|---|---|---|---|---|---|---|---|
| | | | $\mathcal{A}_{Clean}$ | $\mathcal{A}_{\mathbf{PGD}}$ | $\mathcal{A}_{\mathbf{AA}}$ | $\mathcal{A}_{Clean}$ | $\mathcal{A}_{\mathbf{PGD}}$ | $\mathcal{A}_{\mathbf{AA}}$ |
| Tissue | ✗ | Original | 50.61 | 38.50 | 26.96 | 49.73 | 39.30 | 36.95 |
| | | Public (CIFAR-10) | 22.04 | 0.02 | 0.00 | 27.84 | 10.11 | 8.64 |
| | ✓ | **DataFreeShield** | 32.07 | 31.93 | 31.83 | 47.88 | 23.65 | 21.18 |
| Blood | ✗ | Original | 87.95 | 73.37 | 72.31 | 85.64 | 73.13 | 71.99 |
| | | Public (CIFAR-10) | 9.09 | 9.09 | 0.00 | 9.09 | 9.09 | 0.00 |
| | ✓ | **DataFreeShield** | 49.34 | 19.24 | 18.77 | 53.14 | 24.17 | 20.11 |
| Derma | ✗ | Original | 66.90 | 63.90 | 63.01 | 67.58 | 61.99 | 60.14 |
| | | Public (CIFAR-10) | 66.88 | 63.54 | 62.11 | 67.89 | 62.48 | 60.11 |
| | ✓ | **DataFreeShield** | 66.98 | 66.83 | 66.63 | 67.03 | 65.03 | 64.66 |
| OrganC | ✗ | Original | 90.48 | 81.16 | 80.30 | 90.08 | 81.71 | 81.19 |
| | | Public (CIFAR-10) | 79.41 | 40.10 | 36.53 | 84.41 | 46.12 | 43.44 |
| | ✓ | **DataFreeShield** | 76.89 | 46.92 | 45.18 | 82.82 | 53.45 | 51.11 |

Table 13: Comparison of dataset diversification methods on SVHN

| Model | Method | Accuracy | | | Diversity Metric | | | |
|---|---|---|---|---|---|---|---|---|
| | | $\mathcal{A}_{Clean}$ | $\mathcal{A}_{\mathbf{PGD}}$ | $\mathcal{A}_{\mathbf{AA}}$ | Recall ↑ | Coverage ↑ | NDB ↓ | JSD ↓ |
| ResNet-20 | None | 93.31 | 54.11 | 41.03 | 0.801 | 0.230 | 95 | 0.353 |
| | Mixup | 92.13 | **57.71** | 48.17 (+7.14) | 0.882 | 0.241 | **88** | 0.368 |
| | Cutout | 91.34 | 56.01 | **48.29 (+7.26)** | 0.900 | 0.198 | 90 | 0.396 |
| | CutMix | 92.06 | 56.79 | 48.14 (+7.11) | 0.887 | 0.225 | 91 | 0.387 |
| | **DSS (Proposed)** | 91.78 | 54.53 | 45.50 (+4.47) | **0.905** | **0.429** | 90 | **0.237** |
| ResNet-56 | None | 93.17 | 61.40 | 54.38 | 0.821 | 0.218 | **93** | 0.342 |
| | Mixup | 92.23 | 62.26 | 55.11 (+0.73) | 0.848 | 0.226 | **93** | 0.345 |
| | Cutout | 93.92 | 60.54 | 53.80 (-0.58) | 0.842 | 0.164 | 95 | 0.391 |
| | CutMix | 91.20 | 61.46 | 55.38 (-1.00) | 0.871 | 0.189 | 95 | 0.369 |
| | **DSS (Proposed)** | 91.06 | **63.12** | **56.54 (+2.16)** | **0.872** | **0.521** | **93** | **0.154** |
| WRN28-10 | None | 94.26 | 64.94 | 59.99 | 0.246 | 0.147 | 91 | 0.254 |
| | Mixup | 94.50 | 67.51 | 54.70 (-5.29) | 0.252 | 0.120 | 94 | 0.277 |
| | Cutout | 95.51 | 66.77 | 61.96 (+1.97) | 0.305 | 0.060 | 91 | 0.332 |
| | CutMix | 95.67 | 66.71 | 61.16 (+1.17) | 0.321 | 0.100 | 92 | 0.348 |
| | **DSS (Proposed)** | 94.87 | **69.67** | **65.66 (+5.67)** | **0.548** | **0.232** | **88** | **0.190** |

# F    COMPARISON AGAINST TEST-TIME DEFENSE METHODS

Our method DataFreeShield is based on adversarial training (AT), which essentially trains the target model. Although AT has become the dominant approach that is shown to be most effective, there exists test-time defense methods which do not require training of the target model, and instead adopts external detector module or data transformation to mitigate, or cleanse the attacks. To make a fair comparison, we compare our method against two test-time defense methods, DAD Nayak et al. (2022) and TTE Pérez et al. (2021). For implementation, we used the official code provided by the authors. For DAD, we follow their method and retrain a CIFAR-10 pretrained detector on MedMNIST-v2 test set. For TTE, where we used +flip+4crops+4 flipped-crops as it is reported as the best setting in the original paper.

Table 15 shows the results. DAD noticeably does not perform well and there exists a large performance gap to ours in all cases. Note that DAD is evaluated on the same test set that is used to finetuneadapt the detector module, and yet shows poor results. While TTE is advantageous in preserving clean accuracy, the overall robust accuracy is low and not comparable to ours. Both approaches do not directly train the target model, which makes it vulnerable to gradient-based attacks. Thus, comparison against test time defense methods only enhances our assertion that existing methods are insufficient to guarantee robustness when the train data is unavailable.

Table 14: Comparison of dataset diversification methods on CIFAR-10

| Model | Method | Accuracy | | | Diversity Metric | | | |
| | | $\mathcal{A}_{Clean}$ | $\mathcal{A}_{\mathbf{PGD}}$ | $\mathcal{A}_{\mathbf{AA}}$ | Recall ↑ | Coverage ↑ | NDB ↓ | JSD ↓ |
|---|---|---|---|---|---|---|---|---|
| ResNet-20 | None | 82.58 | 23.93 | 14.61 | 0.400 | 0.107 | 88 | 0.355 |
| | Mixup | 84.26 | 16.91 | 5.95 (-8.66) | 0.692 | 0.128 | **87** | 0.372 |
| | Cutout | 82.65 | 26.33 | 17.32 (+2.71) | 0.747 | 0.137 | 95 | 0.369 |
| | CutMix | 83.38 | **28.66** | 18.30 (+3.69) | **0.825** | 0.175 | 89 | 0.347 |
| | **DSS (Proposed)** | 77.83 | 27.42 | **19.09 (+4.48)** | 0.724 | **0.320** | 90 | **0.248** |
| ResNet-56 | None | 83.72 | 30.91 | 27.42 | 0.658 | 0.136 | 93 | 0.310 |
| | Mixup | 83.55 | 32.87 | **27.87 (+0.45)** | 0.761 | 0.135 | 93 | 0.394 |
| | Cutout | 82.96 | 31.39 | 26.83 (-0.59) | 0.853 | 0.113 | 94 | 0.343 |
| | CutMix | 82.60 | 33.78 | 27.86 (+0.44) | **0.892** | 0.150 | 93 | 0.364 |
| | **DSS (Proposed)** | 83.67 | **34.78** | 27.69 (+0.27) | 0.678 | **0.550** | **84** | **0.126** |
| WRN28-10 | None | 91.46 | 43.66 | 36.34 | 0.535 | 0.101 | 91 | 0.253 |
| | Mixup | 90.61 | 48.16 | 36.43 (+0.09) | 0.641 | 0.084 | 94 | 0.322 |
| | Cutout | 92.59 | 39.84 | 34.39 (-1.95) | 0.535 | 0.034 | 95 | 0.443 |
| | CutMix | 91.90 | 42.79 | 34.79 (-1.55) | **0.845** | 0.084 | 93 | 0.328 |
| | **DSS (Proposed)** | 88.16 | **50.13** | **41.40 (+5.06)** | 0.830 | **0.163** | **88** | **0.211** |

Table 15: Comparison on test time defense methods on MedMNISTv2.

| Dataset | Test-Time | Method | ResNet-18 | | |
| | | | $\mathcal{A}_{Clean}$ | $\mathcal{A}_{\mathbf{PGD}}$ | $\mathcal{A}_{\mathbf{AA}}$ |
|---|---|---|---|---|---|
| Tissue | ✓ | DAD | 53.90 | 3.53 | 3.12 |
| | ✓ | TTE | 67.23 | 8.34 | 7.22 |
| | ✗ | **DataFreeShield** | 32.07 | **31.93** | **31.83** |
| Blood | ✓ | DAD | 81.18 | 6.40 | 6.05 |
| | ✓ | TTE | 95.79 | 9.09 | 9.09 |
| | ✗ | **DataFreeShield** | 49.34 | **19.24** | **18.77** |
| Derma | ✓ | DAD | 67.53 | 11.02 | 6.98 |
| | ✓ | TTE | 74.21 | 18.15 | 23.54 |
| | ✗ | **DataFreeShield** | 66.98 | **66.83** | **66.63** |
| OrganC | ✓ | DAD | 72.87 | 38.46 | 34.62 |
| | ✓ | TTE | 87.76 | 36.05 | 35.90 |
| | ✗ | **DataFreeShield** | 76.89 | **46.92** | **45.18** |

Table 16: Evaluation under adaptive attacks on SVHN and CIFAR-10

| Dataset | Model | $\mathcal{A}_{Clean}$ | $\mathcal{A}_{PGD_{CE}}$ | $\mathcal{A}_{AA}$ | Adaptive Attack | | | |
|---------|-------|---------|----------|------|----------|----------|----------|----------|
| | | | | | $\mathcal{A}_{Latent}$ | $\mathcal{A}_{PGD_{(a)}}$ | $\mathcal{A}_{PGD_{(b)}}$ | $\mathcal{A}_{PGD_{(c)}}$ |
| | ResNet-20 | 91.83 | 54.82 | 47.55 | 74.38 | 71.95 | 55.84 | 55.24 |
| SVHN | ResNet-56 | 88.66 | 62.05 | 57.54 | 77.99 | 77.04 | 62.95 | 62.58 |
| | WRN28-10 | 94.14 | 69.60 | 62.66 | 87.68 | 81.39 | 70.64 | 70.08 |
| | ResNet-20 | 74.79 | 29.29 | 22.65 | 65.63 | 54.64 | 31.05 | 30.46 |
| CIFAR-10 | ResNet-56 | 81.30 | 35.55 | 30.51 | 67.73 | 60.61 | 36.94 | 36.43 |
| | WRN28-10 | 86.74 | 51.13 | 43.73 | 79.38 | 70.40 | 51.82 | 51.21 |

Table 17: Evaluation under adaptive attacks on MedMNISTv2

| Dataset | Model | $\mathcal{A}_{Clean}$ | $\mathcal{A}_{PGD_{CE}}$ | $\mathcal{A}_{AA}$ | Adaptive Attack | | | |
|---------|-------|---------|----------|------|----------|----------|----------|----------|
| | | | | | $\mathcal{A}_{Latent}$ | $\mathcal{A}_{PGD_{(a)}}$ | $\mathcal{A}_{PGD_{(b)}}$ | $\mathcal{A}_{PGD_{(c)}}$ |
| Tissue | | 32.07 | 31.93 | 31.83 | 32.07 | 31.98 | 31.95 | 31.95 |
| Blood | ResNet-18 | 49.34 | 19.24 | 18.77 | 20.32 | 29.87 | 24.70 | 24.12 |
| Derma | | 66.98 | 66.83 | 66.63 | 66.98 | 66.53 | 66.98 | 66.93 |
| OrganC | | 76.89 | 46.92 | 45.18 | 76.60 | 65.47 | 48.40 | 48.10 |

## G  EVALUATION UNDER ADAPTIVE ADVERSARIAL ATTACKS

Evaluating robust accuracy using PGD Madry et al. (2018) and AutoAttack Croce & Hein (2020) are considered de facto standard to demonstrate method's robustness. However, we extend our experiments and provide further evaluation under adaptive attacks, including latent attack Sabour et al. (2016) and using different combinations of our training loss $\mathcal{L}_{DFShield}$ as the inner maximization of PGD. Each replaces the coventionally used cross entropy loss $CE(S(x')\|y)$ with: (a) $KL(S(x')\|S(x))$, (b) $KL(S(x')\|T(x))$, (c) $KL(S(x')\|T(x)) + KL(S(x')\|S(x))$. For latent attack Sabour et al. (2016), we followed the original implementation, and used output from the penultimate layer (before flattening), L-BFGS for attack optimization with $\epsilon$=10/255 for perturbation bound. The results are shown in Table 16 and Table 17, where our method DataFreeShield is effective against adaptive attacks as well. In all datasets and models, none of the adaptive attack methods were stronger than cross entropy based PGD and AutoAttack.

## H  FURTHER VISUALIZATION OF LOSS SURFACE

We extend Figure 4 to different models, ResNet-56 and WRN-28-10. The visualization results are shown in Figures 6 and 7. In all visualization settings, applying GradRefine to data-free adversarial training achieves a flatter loss surface. This analysis further supports the experimental results that GradRefine contributes to better performance.

## I  SENSITIVITY ON THE NUMBER OF AGGREGATED BATCHES

In this section, we show sensitivity study on the number of aggregated batches when applying GradRefine. Table 18 shows the performance under varying number of aggregated batches ($\mathcal{B}$) during training. Aggregated Batch being 1 means GradRefine was not applied. For both models, we can observe that the performance is relatively stable for a wide range of $\mathcal{B}$. Also, a smaller model displays slightly higher sensitivity towards $\mathcal{B}$, while a larger model is less affected by it. We found $\mathcal{B} = \{10, 20\}$ to work generally well across different datasets and models.

## J  GENERATED SYNTHETIC SAMPLES

In this section, we display generated synthetic samples used in our experiments, including the baseline methods. The resulting images are displayed in Figure 8 to Figure 17. The overall quality of the baseline samples are noticeably poor, with limited diversity and fidelity. While these images are sufficient for specific tasks such as knowledge distillation or model compression, they are unable

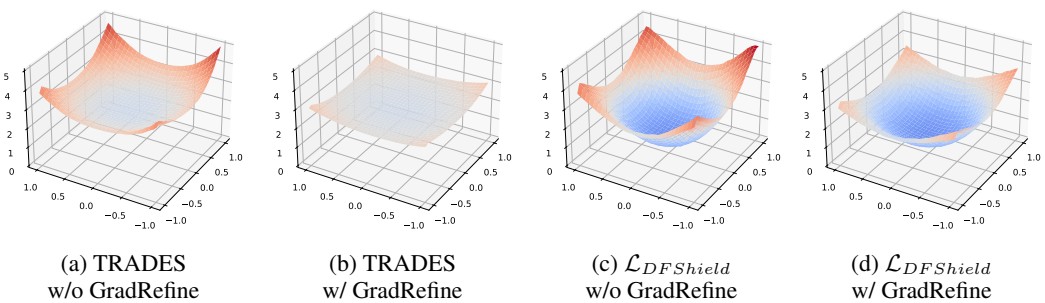

| (a) TRADES | (b) TRADES | (c) $\mathcal{L}_{DFShield}$ | (d) $\mathcal{L}_{DFShield}$ |
| w/o GradRefine | w/ GradRefine | w/o GradRefine | w/ GradRefine |

Figure 6: Loss surface visualization of ResNet56 model trained by data-free AT methods. Each figure represents different training losses with or without GradRefine. We use normalized random direction for $x,y$ axis, following Li et al. (2018). The figures demonstrate that GradRefine achieves flatter loss surfaces.

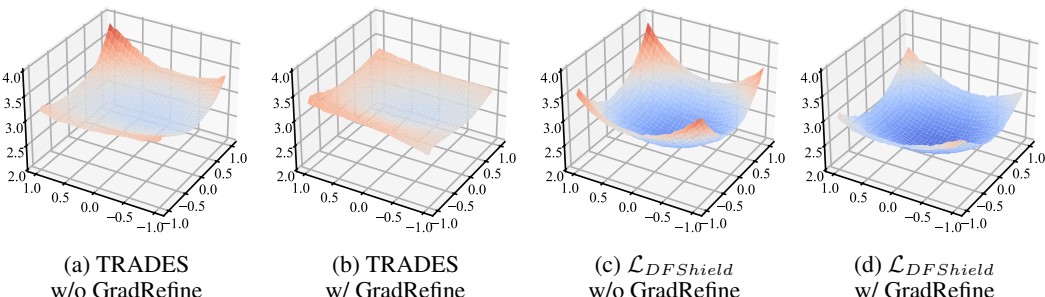

| (a) TRADES | (b) TRADES | (c) $\mathcal{L}_{DFShield}$ | (d) $\mathcal{L}_{DFShield}$ |
| w/o GradRefine | w/ GradRefine | w/o GradRefine | w/ GradRefine |

Figure 7: Loss surface visualization of WRN-28-10 model trained by data-free AT methods. Each figure represents different training losses with or without GradRefine. We use normalized random direction for $x,y$ axis, following Li et al. (2018). The figures demonstrate that GradRefine achieves flatter loss surfaces.

Table 18: Sensitivity study on aggregated batch number using CIFAR-10 dataset.

| $\mathcal{B}$ | ResNet-20 | | | WRN28-10 | | |
|---|---|---|---|---|---|---|
| | $\mathcal{A}_{Clean}$ | $\mathcal{A}_{\mathbf{PGD}}$ | $\mathcal{A}_{\mathbf{AA}}$ | $\mathcal{A}_{Clean}$ | $\mathcal{A}_{\mathbf{PGD}}$ | $\mathcal{A}_{\mathbf{AA}}$ |
| 1 | 77.83 | 27.42 | 19.09 | 88.16 | 50.13 | 41.40 |
| 2 | 75.77 | 29.16 | 22.44 | 88.07 | 50.50 | 41.96 |
| 4 | 74.74 | 29.19 | 22.94 | 87.85 | 50.36 | 42.10 |
| 8 | 75.01 | 29.47 | 23.09 | 87.67 | 50.53 | 41.80 |
| 10 | 75.53 | 29.69 | 22.95 | 87.65 | 50.75 | 42.35 |
| 20 | 74.63 | 29.28 | 22.63 | 86.74 | 51.13 | 43.73 |
| 40 | 28.87 | 13.72 | 10.35 | 85.48 | 50.39 | 44.43 |

to give necessary amount of information needed in robust training. On the other hand, diversified sample synthesis is able to generate diverse samples that are also high in fidelity. For example, in Figure 9 and Figure 10, diversified sample synthesis restores colors and shapes of the original data, while also generating non-overlapping, diversified set of examples. Also, for SVHN, diversified sample synthesis is the only method that is able to generate readable numbers that are recognizable to human eyes. Even in CIFAR-10, a dataset with more complex features, diversified sample synthesis generates samples that faithfully restore the knowledge learned from the original dataset. For larger models with more capacity, the generated samples show recognizable objects such as dogs, airplanes, frogs, etc. The difference in the quality of the generated samples, in addition to the experiment results show that fidelity and diversity of train data play crucial role in robust training.

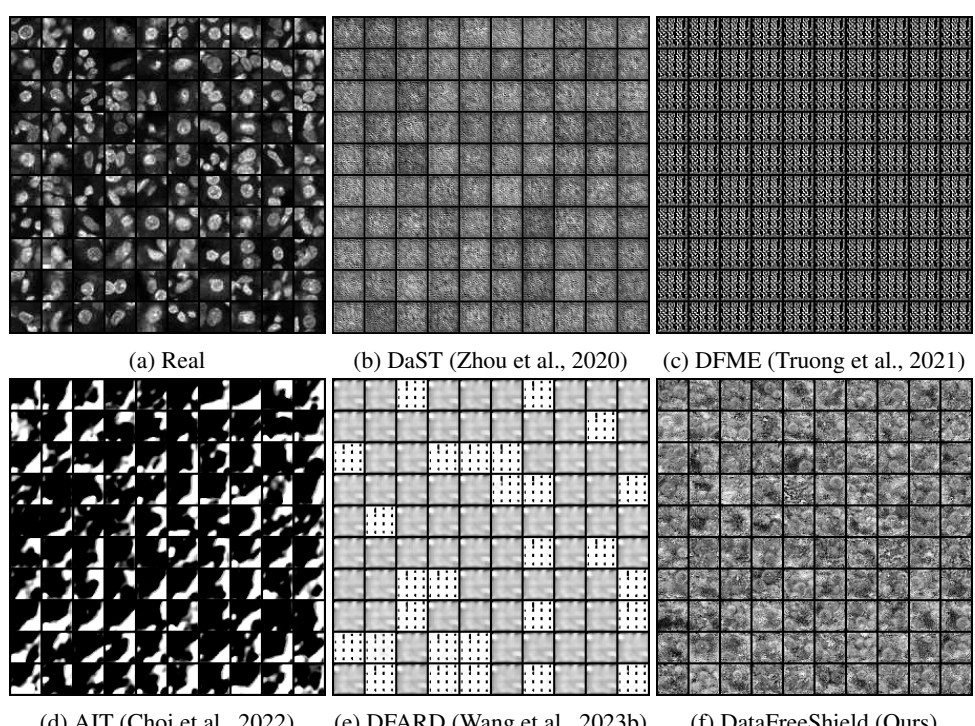

(a) Real      (b) DaST (Zhou et al., 2020)   (c) DFME (Truong et al., 2021)

(d) AIT (Choi et al., 2022)  (e) DFARD (Wang et al., 2023b)  (f) DataFreeShield (Ours)

Figure 8: TissueMNIST

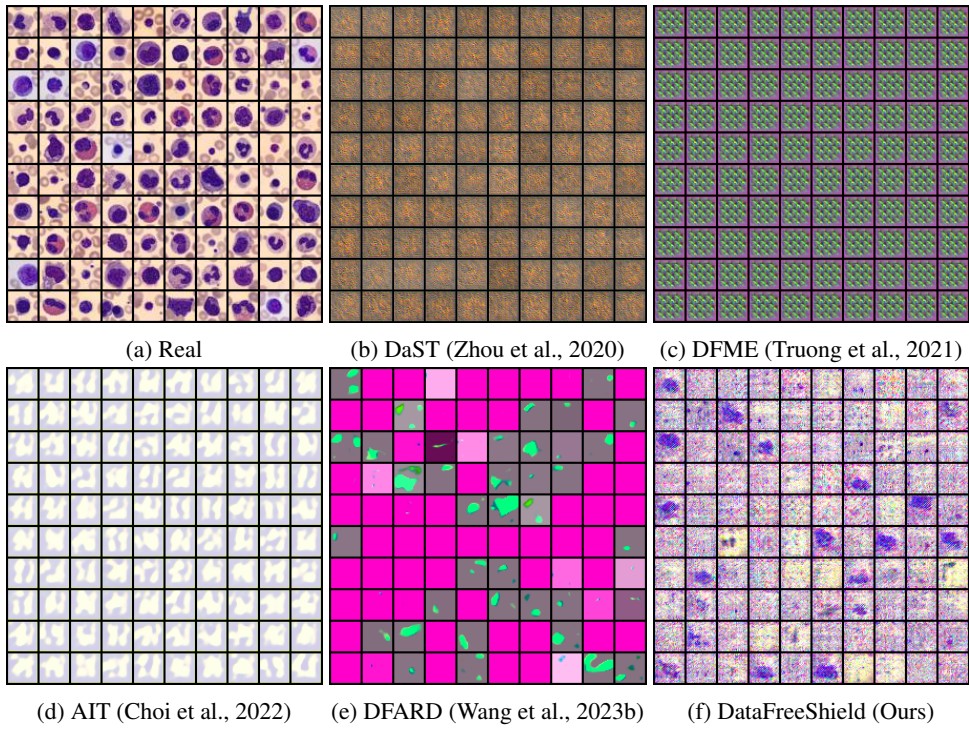

(a) Real                    (b) DaST (Zhou et al., 2020)      (c) DFME (Truong et al., 2021)

(d) AIT (Choi et al., 2022)    (e) DFARD (Wang et al., 2023b)    (f) DataFreeShield (Ours)

Figure 9: BloodMNIST

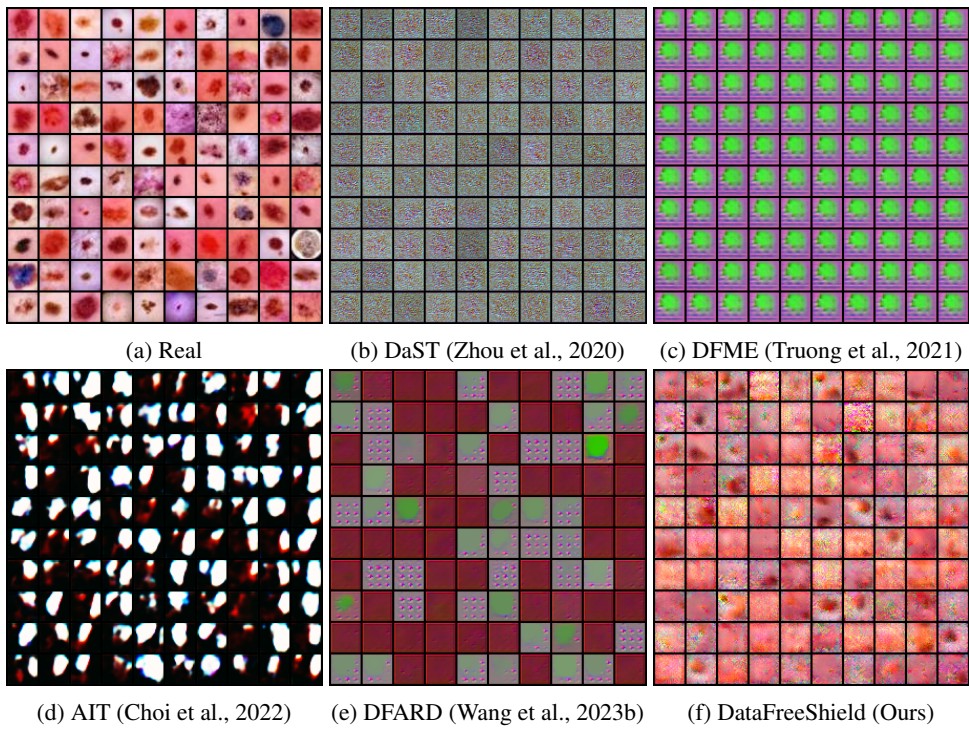

(a) Real                    (b) DaST (Zhou et al., 2020)      (c) DFME (Truong et al., 2021)

(d) AIT (Choi et al., 2022)    (e) DFARD (Wang et al., 2023b)    (f) DataFreeShield (Ours)

Figure 10: DermaMNIST

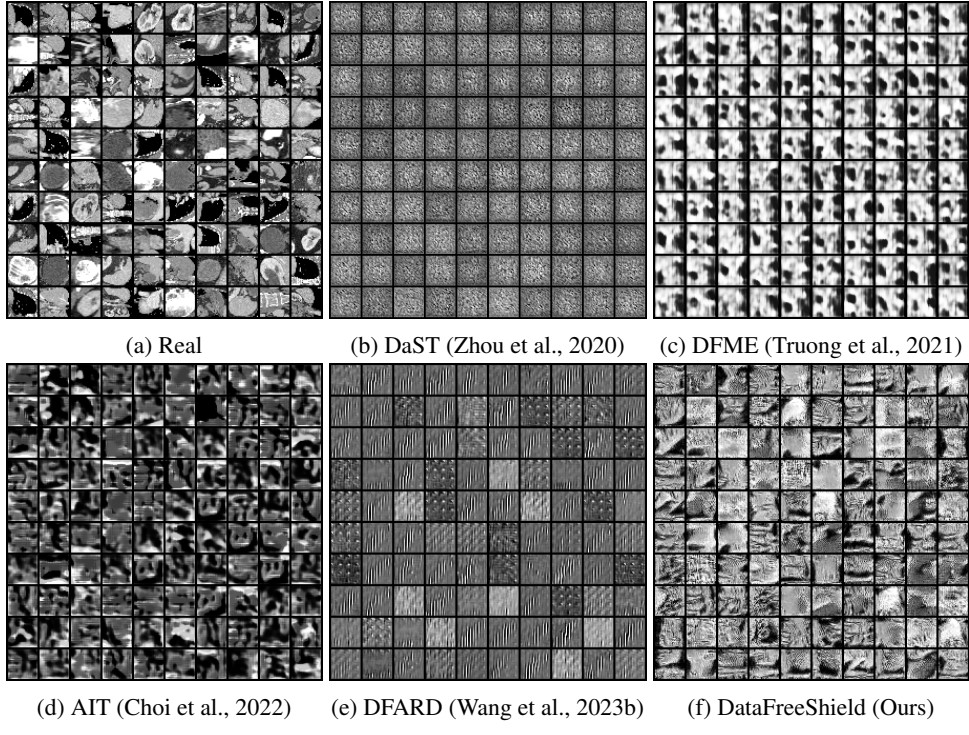

|(a) Real | (b) DaST (Zhou et al., 2020) | (c) DFME (Truong et al., 2021) |

|(d) AIT (Choi et al., 2022) | (e) DFARD (Wang et al., 2023b) | (f) DataFreeShield (Ours) |

Figure 11: OrganCMNIST

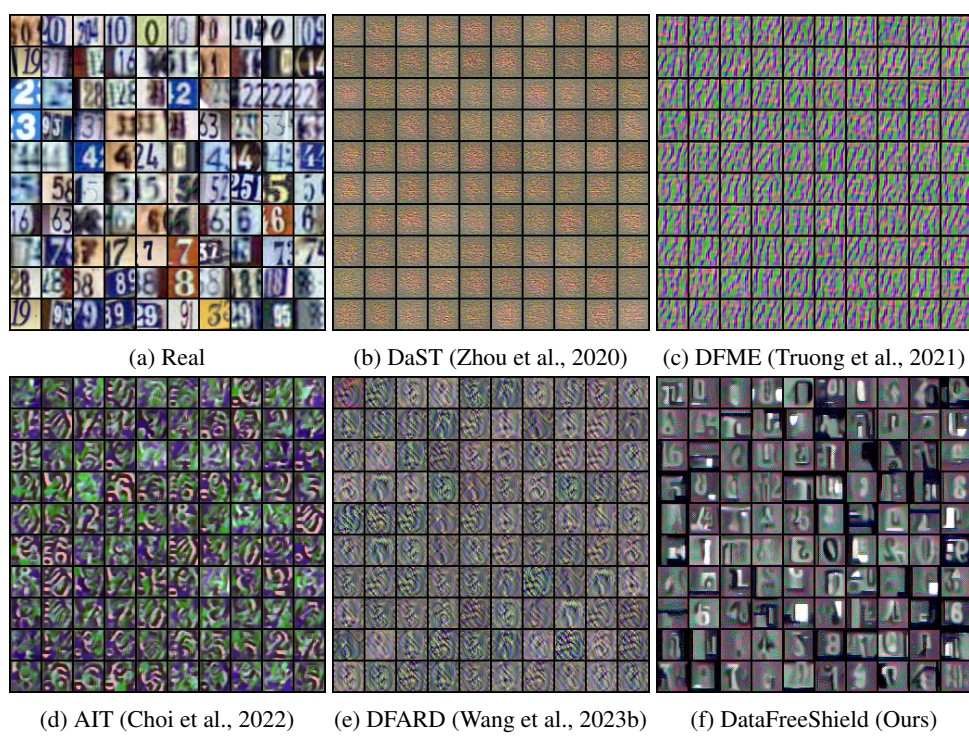

|(a) Real | (b) DaST (Zhou et al., 2020) | (c) DFME (Truong et al., 2021) |

|(d) AIT (Choi et al., 2022) | (e) DFARD (Wang et al., 2023b) | (f) DataFreeShield (Ours) |

Figure 12: SVHN, ResNet-20

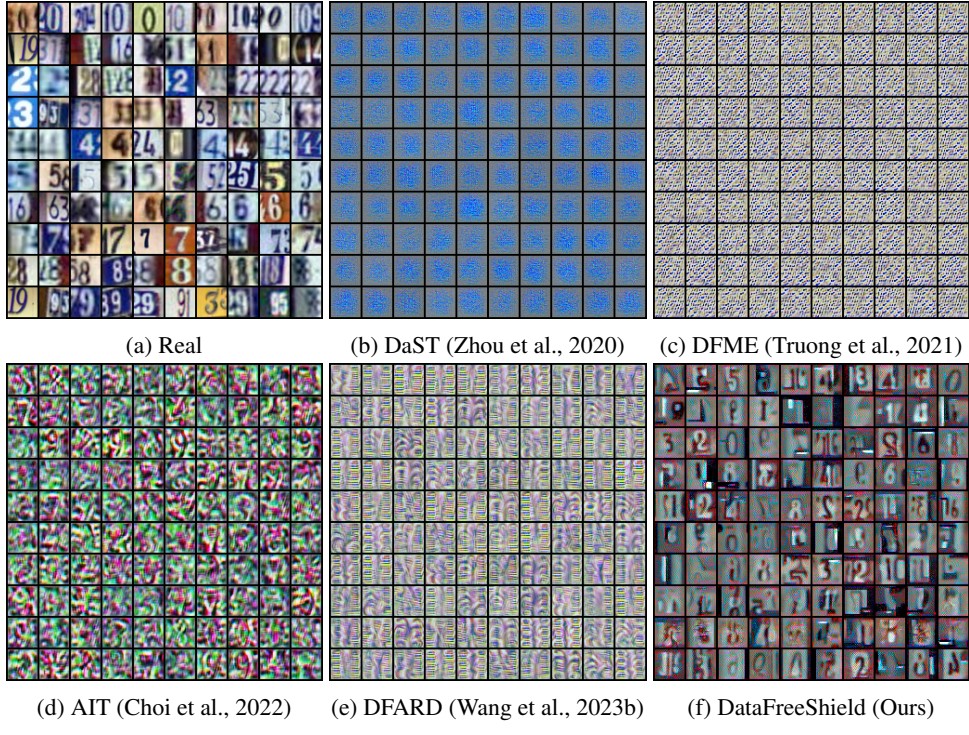

(a) Real       (b) DaST (Zhou et al., 2020)       (c) DFME (Truong et al., 2021)

(d) AIT (Choi et al., 2022)       (e) DFARD (Wang et al., 2023b)       (f) DataFreeShield (Ours)

Figure 13: SVHN, ResNet-56

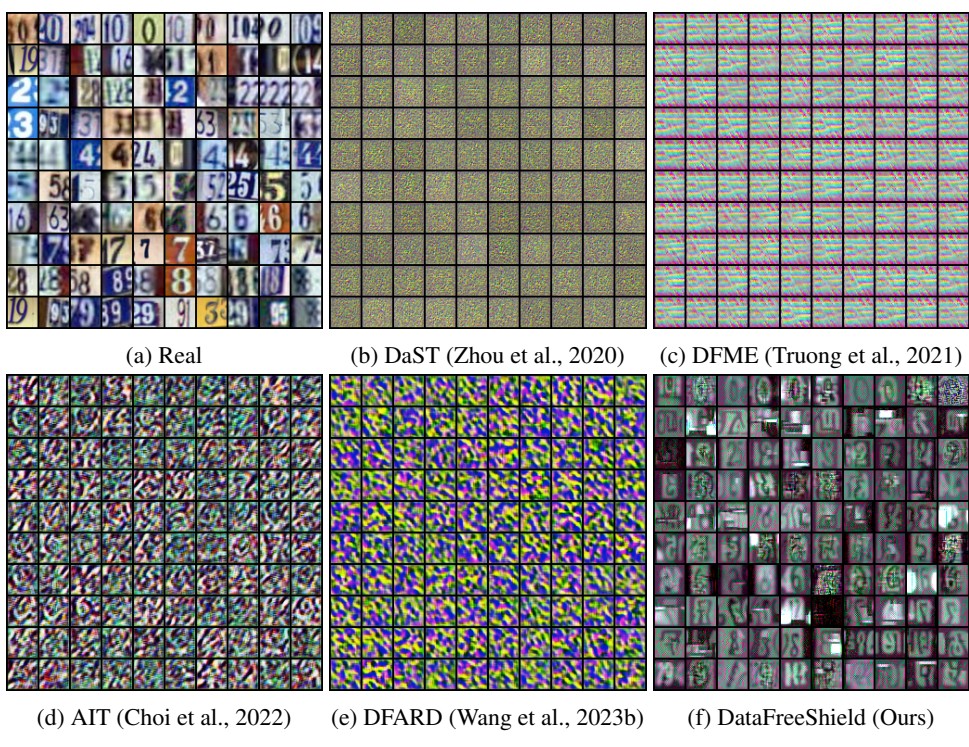

(a) Real       (b) DaST (Zhou et al., 2020)       (c) DFME (Truong et al., 2021)

(d) AIT (Choi et al., 2022)       (e) DFARD (Wang et al., 2023b)       (f) DataFreeShield (Ours)

Figure 14: SVHN, WRN-28-10

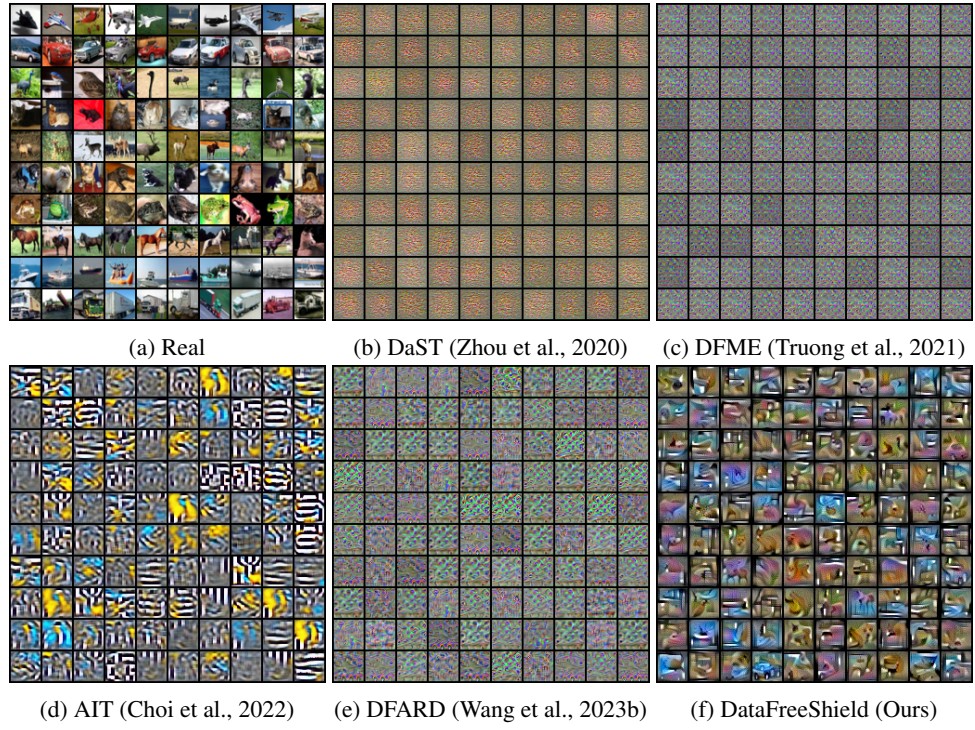

(a) Real      (b) DaST (Zhou et al., 2020)      (c) DFME (Truong et al., 2021)

(d) AIT (Choi et al., 2022)      (e) DFARD (Wang et al., 2023b)      (f) DataFreeShield (Ours)

Figure 15: CIFAR-10, ResNet-20

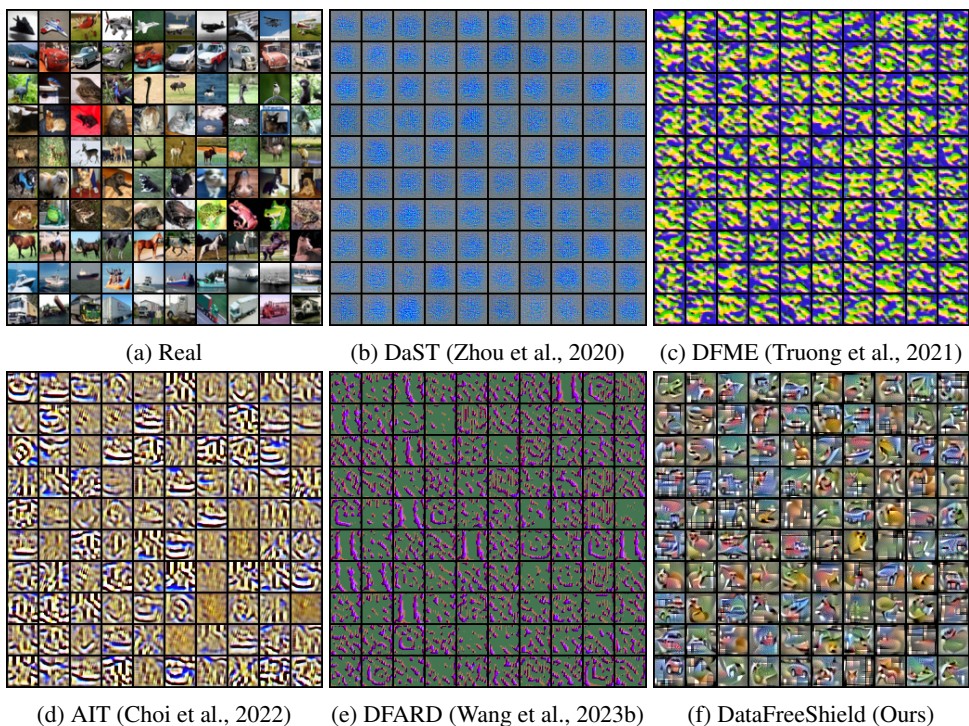

(a) Real      (b) DaST (Zhou et al., 2020)      (c) DFME (Truong et al., 2021)

(d) AIT (Choi et al., 2022)      (e) DFARD (Wang et al., 2023b)      (f) DataFreeShield (Ours)

Figure 16: CIFAR-10, ResNet-56

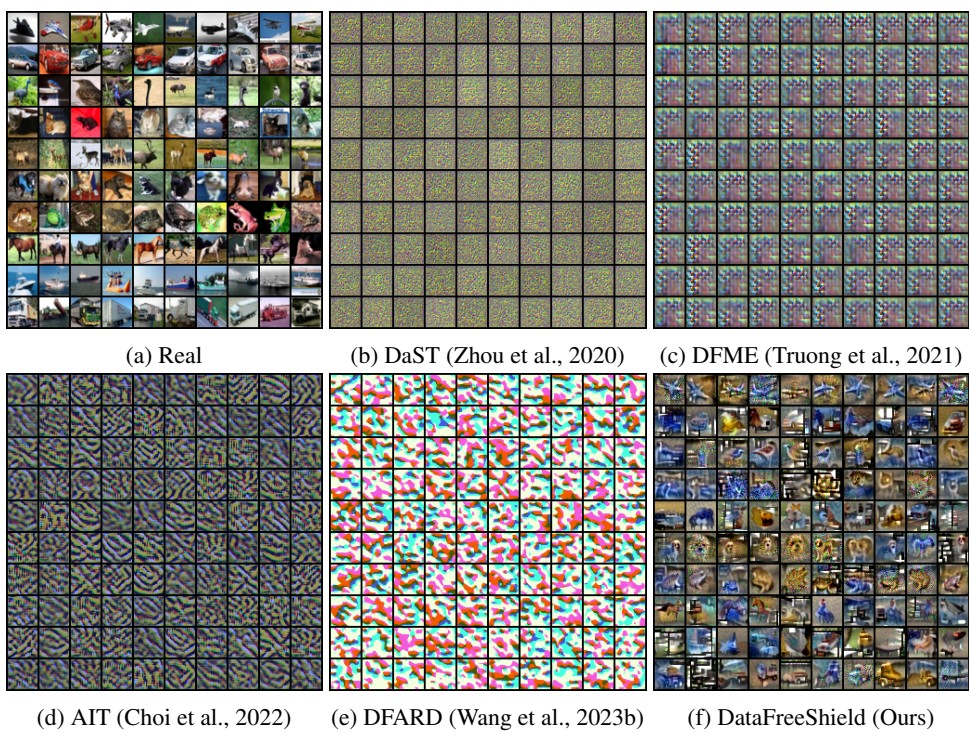

(a) Real      (b) DaST (Zhou et al., 2020)      (c) DFME (Truong et al., 2021)

(d) AIT (Choi et al., 2022)      (e) DFARD (Wang et al., 2023b)      (f) DataFreeShield (Ours)

Figure 17: CIFAR-10, WRN-28-10

