# OpenReview forum: "DataFreeShield: Defending Adversarial Attacks without Training Data"
_ICLR.cc/2024/Conference — Submitted to ICLR 2024_

### Official Review · Reviewer_nfcV · 2023-10-30

**Soundness:** 4 excellent
**Presentation:** 3 good
**Contribution:** 3 good
**Rating:** 5
**Confidence:** 4

**Summary:**

In this work, the authors focus on improving the robustness of deep models without access to the training data. Specifically, given a trained model, the authors first synthesize images and adopt a soft label loss to finetune the model. With such finetuning, the model's robustness can be enhanced.

**Strengths:**

1. The paper is well-written and easy to follow.

2. The problem is very interesting, which enhances the model's robustness without the training data. It might be useful in some cases.

3. The authors have conducted extensive experiments to validate the effectiveness.

**Weaknesses:**

1. It is not clear how you synthesize the samples. Especially, how can you adopt Eq (8) to generate the samples?

2. It is not clear why the authors can adopt synthetic data to improve the model robustness while the images from other domains cannot. In my opinion, the synthetic data is also from different domains.

3. Apart from adversarial training, there are also some defense techniques which does not need training data. For instance, random transforms the input data before feeding them into the model [1]. Also, I am curious if the purifier trained on open datasets, such as ImageNet, can effectively eliminate the adversarial perturbation. There might be other ways that do not need training data to defend against adversarial attacks.


[1] Xie et al. Mitigating Adversarial Effects Through Randomization. ICLR 2018.

**Questions:**

See weakness

---

> ### Author Response · Authors · 2023-11-15
> **Author Response to questions of Reviewer nfcV**
>
> We thank the reviewer for taking the time to review our work. We sincerely address their comments below.
>
> > **It is not clear how you synthesize the samples. Especially, how can you adopt Eq (8) to generate the samples?**
>
> $\to$ In essence, random Gaussian noise is fed to a pretrained teacher, and its output/intermediate values are used to compute the loss terms in Eq (8). Then, the computed loss is backpropagated with respect to the input noise, where the pixels are updated using Adam optimizer. Please also see the red arrows depicted in figure2, and the pseudocode in section B of the appendix for more details.
>
> > **It is not clear why the authors can adopt synthetic data to improve the model robustness while the images from other domains cannot. In my opinion, the synthetic data is also from different domains.**
>
> $\to$ Because synthetic data is always available, while images from other domains are not. To clarify, it is not the matter of whether the data is in-domain or not, but whether the dataset is available or achievable when only a pretrained weight is given. The term “data-free” used in our problem setting, similar to data-free quantization [1-3] and data-free knowledge distillation problems [4,5], means one is only given a pretrained model with no access to its training data. Thus, under this “data-free” constraint, synthetic dataset is always accessible (because we generate them) while other domain dataset is not guaranteed.
>
> [1] Nagel, M. et al. Data-free quantization through weight equalization and bias correction. In Proceedings of the IEEE/CVF International Conference on Computer Vision, 2019.
> [2] Cai, Y. et al. ZeroQ: A novel zero shot quantization framework. In Proceedings of the IEEE/CVF Conference on Computer Vision and Pattern Recognition, 2020.
> [3] Xu, S. et al. Generative low- bitwidth data free quantization. In European Conference on Computer Vision, 2020.
> [4] Nayak, G.K. et al. "Zero-Shot Knowledge Distillation in Deep Networks. In International Conference on Machine Learning, 2019.
> [5] Chen, H. et al. DAFL:Data-Free Learning of Student Networks. In Proceedings of the IEEE International Conference on Computer Vision, 2019.
>
> > **There might be other ways that do not need training data to defend against adversarial attacks.**
>
> $\to$ It is important to note that while different adversarial defense techniques have been suggested in the past, only adversarial training(AT) and its variants have survived. Others have been shown to be ineffective or weak, often unable to defend against strong attacks. For example, multiple quantization and randomization techniques have been proposed as viable defense methods, but were quickly deprived of their efficacy and found to be a false sense of defense that can easily be circumvented [1]. For now, AT has become the dominant approach that is shown to be most effective. Thus we adopt the same AT approach in our problem.
> To support our argument, we provide comparison against test time defense methods, DAD and TTE, in the table below. Notice how they are not competitive against ours in all cases. We will add the following results to Table2 in the revised manuscript.
>
> | Dataset | Method | Clean(%) | PGD(%) | AutoAttack(%) |
> |:-------:|:------:|:--------:|:------:|:-------------:|
> |  Tissue |   DAD  |    53.90 |   3.53 |          3.12 |
> |         |   TTE  |    67.23 |   8.34 |          7.22 |
> |         |  **Ours**  |    32.07 |  31.93 |         31.83 |
> |  Blood  |   DAD  |    81.18 |   6.40 |          6.05 |
> |         |   TTE  |    95.79 |   9.09 |          9.09 |
> |         |  **Ours**  |    49.34 |  19.24 |         18.77 |
> |  Derma  |   DAD  |    67.53 |  11.02 |          6.98 |
> |         |   TTE  |    74.21 |  18.15 |         23.54 |
> |         |  **Ours**  |    66.98 |  66.83 |         66.63 |
> |  OrganC |   DAD  |    72.87 |  38.46 |         34.62 |
> |         |   TTE  |    87.76 |  36.05 |         35.90 |
> |         |  **Ours**  |    76.89 |  46.92 |         45.18 |
>
> [1] Athalye, Anish, Nicholas Carlini, and David Wagner. "Obfuscated gradients give a false sense of security: Circumventing defenses to adversarial examples." International conference on machine learning, 2018.

---

> > ### Author Response · Authors · 2023-11-22
> > **Discussion Reminder**
> >
> > Thank you again for the feedback on our paper. We hope that our responses have addressed your inquiries and concerns. If this is not the case, please inform us and we would be glad to engage in further discussion.

---

### Official Review · Reviewer_F6PX · 2023-10-31

**Soundness:** 2 fair
**Presentation:** 3 good
**Contribution:** 2 fair
**Rating:** 3
**Confidence:** 4

**Summary:**

The paper proposes a data-free defense against adversarial (evasion) attacks for image classifiers. In cases where availability of the training data of the target model is unavailable, authors propose generating an auxillary dataset composed of synthetic images. These synthetic images are generated using the information captured within the pretrained model, and generated in a way that maximizes diversity. The paper further proposes a training method to make the most out of this synthetic data, which uses the soft predictions of the target model as supervision. This training method enforces the model to learn a flat loss landscape, which promotes adversarial robustness.

**Strengths:**

1. Authors do a great job explaining their method, including qualitative results/visualizations wherever possible to convey important points regarding their method. For example, I particularly liked the usage of a toy example to demonstrate how using dynamic loss weights during synthesis process increases diversity of synthetic data. Overall, the writing quality and presentation is great.
2. In my opinion, the methods proposed by authors are simple, intuitive, and effective. These methods include
    - the dynamic loss weighting for synthesis loss to increase diversity,
    - using a regularization term (term 3 in L_DFSshield) to encourage flatness of loss surface, and
    - using a refinement strategy on top of gradients to further enforce flatness of loss surface.

**Weaknesses:**

1. The explanation of the gradient refinement strategy seems rushed. The authors say that targeting a smoother loss landscape will make it more likely that there is alignment between minima achieved using real and synthetic data. Then, the authors describe the design of their gradient refinement strategy. There is no connection presented between the initial idea (smoother loss landscape) and the proposed method to implement/enforce said idea (gradient refinement). As a result, it is not clear how the gradient refinement strategy will lead to smoother loss landscape.
2. Furthermore, there is already a term in the training objective (3rd term) that enforces smoother loss landscape. Then, how does the gradient refinement strategy and the third loss term differ in what they are trying to achieve? The distinciton between the two is not clear.
3. Discussion in the results section is not thorough. Authors provide speculative reasoning behind certain observations, without any support in the form of theoretical/empirical results or references to relevant prior works. In my opinion, such speculative reasoning does not add much value to the paper. For example, in section 5.3. (larger datasets), authors comment that prior works are unable to take advantange of large model capacity due to lack of diversity in synthetic samples generated by them. There are several ways this could have been backed up with results. Off the top of my head, here are few methods: (1) using tSNE plots; (2) performing k-means clustering and measuring sum of squared residuals within clusters; (3) fitting a GMM to the synthetic data generated by different methods and comparing the resulting covariance matrices. I strongly suggest the authors to provide supporting evidence to any speculative claims in the results section.
4. There are several glaring issues in Table 3, that makes it hard to trust the numbers presented in this table.
    a. SVHN/DFARD/ResNet-20: A_pgd (weaker attack) is lower than A_aa (stronger attack)
    b. SVHN/DaST/ResNet-50: A_pgd is lower than A_aa
    c. Cifar10/DFARD/WRN28-10: A_clean is lower than A_pgd
5. The authors claim that their work is the "first data-free adversarial defense". This is clearly not true as one of the papers that authors cite (Nayak, 2022) proposes a similar method, ie DAD. The authors acknowledge that the only difference between their method and DAD is that DAD assumes availability of a an auxiliary dataset which is from the same domain as the training dataset of the target model. Irrespective, DAD does not require access to the original dataset, making it a data-free defense.
6. Continuing from the previous point, authors do not compare their method with DAD. I understand the benefit of adapting other data-free methods to the adversarial training regime and comparing against them. But this is not more useful than comparing against a method that already tackles the exact same problem as the one studied in this paper (ie, no adaptation needed). This comparison is crucial for the paper.
7. Another method that provides a data-free way for improving adversarial defense is the TTE method [b]. The authors neither cite this method nor compare against it. Overall, it is critical to include DAD and TTE as baselines.
8. Authors do not properly explore adaptive attacks. If a defense flattens/smoothens the loss landscape, it makes gradient based attacks harder to converge (and as result, less effective). AutoAttack circumvents this issue to some extent by including a black-box attack, but this is not enough to establish the true robustness of such defenses in my opinion. Using a latent space attack [a] that doesn't rely on the (flattened) outputs of the final layer will be a more effective attack than something like PGD.
9. Continuing my point regarding adaptive attacks, there are no results using an attack that targets all the loss terms used during training collectively. Based on my understanding, the authors perform attacks using the cross-entropy loss only. Performing an exploration regarding how the attack effectiveness changes using different combinations of the training loss terms is important in terms of developing an adaptive attack.

**References**

[a] Sabour, S., Cao, Y., Faghri, F., and Fleet, D. J. Adversarial manipulation of deep representations. International Conference on Learning Representations, 2016.

[b] Pérez, Juan C., et al. "Enhancing adversarial robustness via test-time transformation ensembling." Proceedings of the IEEE/CVF International Conference on Computer Vision. 2021.

**Questions:**

1. In case of pre-training with Derma, why is adversarial training with organC better than doing so with Derma itself (figure 1, right)? Authors comment that this occurs in rare cases without any further explanation. However 1 out of 4 is not rare. Where is this conclusion coming from? Are there additional datapoints including dataset combinations other than the ones used in the paper? What can be the reason behind this phenomenon?
2. In figure 3, for fixed (b) and dynamic (c) coefficient methods, how often do blue points appear in red space and vice versa? If this occurs more often in (c) than (b) due to increased diversity in (c), wouldn't this make it harder for the classifier to learn discriminative features using points generated with (c)? Implying that diversification is counter productive?
3. In Table 2, for Derma => DFME, why are the A_clean and A_pgd numbers the same?
4. Can you please explain the issues in Table 3 (listed in Weaknesses section)?
5. How does the proposed defense compare against DAD and TTE?
6. How does the proposed defense fare against adaptive attacks (see description in Weaknesses section)?

---

> ### Author Response · Authors · 2023-11-15
> **Author Response to questions of Reviewer F6PX (1/2)**
>
> We thank the reviewer for a thorough and detailed review of our work, and the encouraging remarks on the effectiveness of our methods, along with its presentation. We address the concerns and questions provided by the reviewer.
>
> > **DAD does not require access to the original dataset, making it a data-free defense. How does the proposed defense compare against DAD and TTE?**
>
> $\to$ We respectfully argue that both DAD and TTE are not ‘data-free’ for the following reasons. DAD heavily relies on the existence of similar dataset (which we think is already not data-free), and it also uses the **original test data** to tune their model, which is a far extreme assumption and could be regarded as data leak. Also, it is difficult to view TTE as “data-free” because they propose TTE as an enhancement method applied to already robust models trained using the original data.
>
> Despite the unfairness, we compare DAD and TTE with ours. DAD does not perform well and there exists a large performance gap to ours in all cases. Note that DAD is evaluated on the same test set that is used to finetune/adapt the detector module, and yet shows poor results. While TTE is advantageous in preserving clean accuracy, the overall robust accuracy is low and not comparable to ours. Both approaches do not directly train the target model, which makes it vulnerable to gradient-based attacks. Thus, comparison against test time defense methods only enhances our assertion that existing methods are insufficient to guarantee robustness when the train data is unavailable.
>
> We used the official code provided by the authors. For DAD, we follow their method and retrain a Cifar-10 pretrained detector on MedMNIST-v2 test set. For TTE, where we used +flip+4crops+4 flipped-crops as it is reported as the best setting in the original paper. We will add this comparison to Table 2  in the revision.
>
> | Dataset | Method | Clean(%) | PGD(%) | AutoAttack(%) |
> |:-------:|:------:|:--------:|:------:|:-------------:|
> |  Tissue |   DAD  |    53.90 |   3.53 |          3.12 |
> |         |   TTE  |    67.23 |   8.34 |          7.22 |
> |         |  **Ours**  |    32.07 |  31.93 |         31.83 |
> |  Blood  |   DAD  |    81.18 |   6.40 |          6.05 |
> |         |   TTE  |    95.79 |   9.09 |          9.09 |
> |         |  **Ours**  |    49.34 |  19.24 |         18.77 |
> |  Derma  |   DAD  |    67.53 |  11.02 |          6.98 |
> |         |   TTE  |    74.21 |  18.15 |         23.54 |
> |         |  **Ours**  |    66.98 |  66.83 |         66.63 |
> |  OrganC |   DAD  |    72.87 |  38.46 |         34.62 |
> |         |   TTE  |    87.76 |  36.05 |         35.90 |
> |         |  **Ours**  |    76.89 |  46.92 |         45.18 |
>
> > **In case of pre-training with Derma, why is adversarial training with organC better than doing so with Derma itself (figure 1, right)?**
>
> $\to$ Because OrganC dataset contains more train samples than Derma. For MedMNIST-v2, each dataset in the collection contains a different number of training samples. Derma provides 7,007 training samples while OrganC provides 13,940, which is nearly double. This is possibly the reason OrganC provides a slightly better performance than using Derma itself, due to having more adversarial samples to learn robustness from. Also, due to the nature of biomedical dataset, the environment in which the dataset was collected could differ between each dataset, leading to difference in quality of the collected samples. The “rare cases'' we say in our paper refer to these external causes that we usually do not have control over.
>
> > **In figure 3, for fixed (b) and dynamic (c) coefficient methods, how often do blue points appear in red space and vice versa?**
>
> $\to$ (b) shows 2.8% and 6.1% misclassified points for each class, while (c) has 5.6% and 9.1%. The table below shows the exact numbers. For each plot we generated 2048 data points, where half is given label 0 (red) and the other half 1 (blue) when generating with cross entropy loss. So ideally, generated data points should have equal distribution in both regions of color.
>
> |          Method         |  Blues in Red  |   Reds in Blue  |
> |:-----------------------:|:--------------:|:---------------:|
> |  fixed coefficient (b)  | 29/1024 (2.8%) | 62/1024  (6.1%) |
> | dynamic coefficient (c) | 58/1024 (5.6%) | 93/1024  (9.1%) |
>
> In the results, we can see that (c) does show higher error, with 2.8\% and 3.0\% increase in each case. However, the difference is not large enough where it can hinder the model’s ability to learn discriminative features. On the other hand, the gained diversity is highly noticeable. The trade-off with discriminative features is not a counter productive factor but something we can leverage, where a small sacrifice in discriminative features leads to huge improvement in diversity.

---

> ### Author Response · Authors · 2023-11-15
> **Author Response to questions of Reviewer F6PX (2/2)**
>
> > **In Table 2, for Derma => DFME, why are the A_clean and A_pgd numbers the same?**
>
> $\to$ Because most datasets in MedMNIST-v2 have uneven class distributions, and training could easily diverge to one of the classes. In the case of Derma, the class imbalance is severe, where each class contains [66, 103, 220, 23, 223, 1341, 29] samples, 2005 in total. The identical numbers reported in the table is due to the model failing to learn and simply diverging in random guessing one of these classes. For example, if the model diverges towards 5th class, the resulting accuracy is 11.12\%(=223/2005), which is why ResNet-18 trained using DFME shows this number. And since the model has diverged, all the gradient-based attacks become ineffective and thus the numbers are the same across all settings (clean, PGD, AutoAttack). Thus the results only mean that DFME makes training unstable and difficult to converge when applied to our problem. To make a fair observation, we will revise the table to report F1 scores instead of naive accuracy. We hope this alleviates the reviewer’s concern towards the credibility of the experiment results.
>
> > **Can you please explain the issues in Table 3 (listed in Weaknesses section)?**
>
> a. SVHN/DFARD/ResNet-20: A_pgd (weaker attack) is lower than A_aa (stronger attack).
> b. SVHN/DaST/ResNet-56: A_pgd is lower than A_aa.
> c. Cifar10/DFARD/WRN28-10: A_clean is lower than A_pgd.
>
> $\to$ a & c: are possible outcomes of synthetic data training, and possible signs of gradient obfuscation[1]. The signs of gradient obfuscation include increasing attack strength not necessarily leading to better attack success rate, which is what we can observe from Table 3. We agree the phenomenon is rarely observed in real-data adversarial training, where the train dataset is reliable and does not contain any outliers in terms of distribution or quality. However, all the baseline methods are data-free (See Appendix A.4 for explanation on baseline methods and how we adapt them to our problem), and use synthetic data for training. Learning from these data is tricky and adding adversarial perturbation only makes it harder. Thus, the model often diverges, or simply learns to circumvent gradient-based attacks in an unintended way.
>
> $\to$ b: We found that for SVHN ResNet56 DaST, there has been an error, where 20.20 was written as 0.20. We apologize for the mistake and causing confusion to the reviewers.
>
> [1] Athalye, Anish, Nicholas Carlini, and David Wagner. "Obfuscated gradients give a false sense of security: Circumventing defenses to adversarial examples." International conference on machine learning, 2018.
>
> > **How does the proposed defense fare against adaptive attacks?**
>
> $\to$  We are to update the results using adaptive attacks.
>
> >  **It is not clear how the gradient refinement strategy will lead to smoother loss landscape. Also, how does the gradient refinement strategy and the third loss term differ in what they are trying to achieve?**
>
> $\to$  The two methods essentially hold the same objective, but achieves it in a distinct way. Their ultimate goal is to learn adversarial robustness from synthetic samples that can be applied to real attacks. The third term in the training loss penalizes the model when it is highly sensitive to small perturbations. Implicitly, this would make the model favor a set of parameters that have a smoother loss surface with respect to the input. On the other hand, the gradient refinement method explicitly ignores gradients of parameters that are highly fluctuating, pushing the model towards a smoother loss landscape where loss does not fluctuate with small changes in the parameters. The ablation study in Table7 shows that improvement from both the loss term and the gradient refinement method are distinct and best when used together.

---

> > ### Author Response · Authors · 2023-11-19
> > **Author Response to Reviewer F6PX (Update on additional experiment)**
> >
> > > **How does the proposed defense fare against adaptive attacks?**
> >
> > $\to$ Our method is effective against adaptive attacks as well. As the table below shows, none of the adaptive attack methods were stronger than PGD(CE) and AutoAttack. As far as we know, showing robust accuracy under PGD(CE) and AutoAttack are considered de facto standard to demonstrate method’s robustness [1-5].
> > As the reviewer requested, we compared latent space attack [6] and three different combinations of the training loss. Notice that none of the adaptive attack methods fare against AutoAttack and PGD.
> >
> > |  Dataset |   Model   | Clean | PGD(CE(S(x'),y) | AutoAttack | latent [6] | PGD(KL(S(x')\|\|S(x)) | PGD(KL(S(x')\|\|T(x)) | PGD(KL(S(x')\|\|T(x)) + KL(S(x')\|\|S(x))) |
> > |:--------:|:---------:|:-----:|-----------------|------------|--------|-----------------------|-----------------------|--------------------------------------------|
> > |   SVHN   | ResNet-20 | 91.83 | 54.82           | 47.55      | 74.38  | 71.95                 | 55.84                 | 55.24                                      |
> > |          | ResNet-56 | 88.66 | 62.05           | 57.54      | 77.99  | 77.04                 | 62.95                 | 62.58                                      |
> > |          | WRN28-10  | 94.14 | 69.60           | 62.66      | 87.68  | 81.39                 | 70.64                 | 70.08                                      |
> > | Cifar-10 | ResNet-20 | 74.79 | 29.29           | 22.65      | 65.63  | 54.64                 | 31.05                 | 30.46                                      |
> > |          | ResNet-56 | 81.30 | 35.55           | 30.51      | 67.73  | 60.61                 | 36.94                 | 36.43                                      |
> > |          | WRN28-10  | 86.74 | 51.13           | 43.73      | 79.38  | 70.40                 | 51.82                 | 51.21                                      |
> >
> > | Dataset |   Model   | Clean | PGD(CE(S(x'),y) | AutoAttack | latent [6] | PGD(KL(S(x')\|\|T(x)) | PGD(KL(S(x')\|\|S(x)) | PGD(KL(S(x')\|\|T(x)) + KL(S(x')\|\|S(x))) |
> > |:-------:|:---------:|:-----:|-----------------|------------|-------------|-----------------------|-----------------------|--------------------------------------------|
> > |  Tissue | ResNet-18 | 32.07 | 31.93           | 31.83      | 32.07       | 31.98                 | 31.95                 | 31.95                                      |
> > |  Blood  |           | 49.34 | 19.24           | 18.77      | 20.32       | 29.87                 | 24.70                 | 24.12                                      |
> > | Derma   |           | 66.98 | 66.83           | 66.63      | 66.98       | 66.53                 | 66.98                 | 66.93                                      |
> > | OrganC  |           | 76.89 | 46.92           | 45.18      | 76.60       | 65.47                 | 48.40                 | 48.10                                      |
> >
> > For [6], we followed the original implementation, and used output from the penultimate layer (before flattening), L-BFGS for attack optimization with $\epsilon$=10/255 for perturbation bound.
> >
> > [1] Wang, Zekai, et al. "Better diffusion models further improve adversarial training." International Conference on Machine Learning, 2023.
> > [2] Xu, Yuancheng, et al. "Exploring and Exploiting Decision Boundary Dynamics for Adversarial Robustness." International Conference on Learning Representations, 2023.
> > [3] Gowal, Sven, et al. "Improving robustness using generated data." Advances in Neural Information Processing Systems, 2021.
> > [4] Pang, Tianyu, et al. "Bag of Tricks for Adversarial Training." International Conference on Learning Representations, 2020.
> > [5] Sehwag, Vikash, et al. "Robust Learning Meets Generative Models: Can Proxy Distributions Improve Adversarial Robustness?." International Conference on Learning Representations. 2021.
> > [6] Sabour, S., Cao, Y., Faghri, F., and Fleet, D. J. Adversarial manipulation of deep representations. International Conference on Learning Representations, 2016.

---

> > > ### Author Response · Authors · 2023-11-22
> > > **Discussion Reminder**
> > >
> > > Thank you again for the feedback on our paper. We hope that our responses have addressed your inquiries and concerns. If this is not the case, please inform us and we would be glad to engage in further discussion.

---

### Official Review · Reviewer_RMmq · 2023-10-31

**Soundness:** 2 fair
**Presentation:** 2 fair
**Contribution:** 2 fair
**Rating:** 5
**Confidence:** 4

**Summary:**

This paper proposes a data-free adversarial training method, DataFreeShield, which creates a synthetic dataset and performs adversarial training on the synthetic dataset to obtain a robust model. Experiments show that the proposed method outperforms several baselines.

**Strengths:**

1. This paper explores a novel, realistic scenario-based approach to adversarial training.
2. The motivation of the entire framework is clear.
3. This paper is richly designed with experiments.

**Weaknesses:**

1. Although this setup is interesting, we have doubts about its actual performance. Compared with the adversarial training model, the adversarial robustness obtained in Table 3 is very low and difficult to use in practice. Especially on CIFAR100, the robust accuracy of ResNet-20 under AA is only 5.97, and the clean accuracy is significantly lower than the normal model (60%+).

2. Please analyze the time complexity of the compared methods, which is important for practical applications.

3. How to extract robust knowledge from clean images is not clearly expressed in this paper. The adversarial robustness of previous work relies on pre-trained robust models, but why can adversarial robustness be obtained using only loss constraints? If this is the case, can it be used in any adversarial training? We believe that this section needs to be described in detail.

4. What is the relationship between adversarial robustness and the amount of generated data?

**Questions:**

See Weaknesses.

---

> ### Author Response · Authors · 2023-11-15
> **Author Response to questions of Reviewer RMmq (1/2)**
>
> We thank reviewer RMmq for the encouraging remarks on novelty and the presentation of our work. We address your concerns as follows.
>
> > **Compared with the adversarial training model, the adversarial robustness obtained in Table 3 is very low and difficult to use in practice.**
>
> $\to$ We respectfully disagree with the reviewer’s comment on the amount of robustness and the practicality of the proposal. The reviewer compares our results against the conventional approach where the original train set is available. We gently remind the reviewer that the purpose of studying this particular setup is not to propose data-free approach’s superiority against data-driven ones, but to provide an alternative solution when none is available. (And currently, there is no known solution to make a neural network robust without using the original dataset.)
> We provide a few examples from other tasks to show that data-free methods have an inevitable gap with data-driven ones. For knowledge distillation, data-free approach[1] compares up to -10% to the data-driven one by Hinton[6]. This gap is larger for quantization, where data-free approaches show degradation up to -42.67% compared to data-driven quantization work[7] of the same year.
>
> **Table 1 : Data-free vs Data-driven in Knowledge Distillation**
> | | | |  Data-driven | Data-free | |
> |-----------|--------------|-------------|-------------|--------|---|
> | **Dataset** | **Model**| **Teacher Acc.** | **KD (2015)** | **ZSKD (2019)** | **Perf. Gap** |
> | CIFAR-10 | AlexNet | 83.03 | 80.08 | 69.56 | ***-10.52*** |
>
>
> **Table 2 : Data-free vs Data-driven in 4-bit Quantization**
> |             |           |                  | Data-driven         | Data-free      |                  |                 |
> |-------------|-----------|------------------|---------------------|----------------|------------------|-----------------|
> | **Dataset** | **Model** | **Teacher Acc.** | **Adaround (2020)** | **DFQ (2019)** | **ZeroQ (2020)** | **GDFQ (2020)** |
> | ImageNet    | ResNet-18 | 69.68            | 68.71               | 38.98          | 26.04            | 60.60           |
> |             |           |                  | **Perf. Gap**       | ***-29.73***   | ***-42.67***     | ***-8.11***     |
>
>
>
> [1] Nayak, G.K. et al. "Zero-Shot Knowledge Distillation in Deep Networks. In International Conference on Machine Learning, 2019.
> [2] Chen, H. et al. DAFL:Data-Free Learning of Student Networks. In Proceedings of the IEEE International Conference on Computer Vision, 2019.
> [3] Nagel, M. et al. Data-free quantization through weight equalization and bias correction. In Proceedings of the IEEE/CVF International Conference on Computer Vision, 2019.
> [4] Cai, Y. et al. ZeroQ: A novel zero shot quantization framework. In Proceedings of the IEEE/CVF Conference on Computer Vision and Pattern Recognition, 2020.
> [5] Xu, S. et al. Generative low- bitwidth data free quantization. In European Conference on Computer Vision, 2020.
> [6] Hinton, G. et al. Distilling the Knowledge in a Neural Network. In Advances in Neural Information Processing Systems, Deep Learning Workshop, 2015.
> [7] Nagel, M. et al. Up or Down? Adaptive Rounding for Post-Training Quantization. In International Conference on Machine Learning, 2020.
>
> > **Please analyze the time complexity of the compared methods.**
>
> $\to$ Time complexity for our method is O(N*(F+B)) for data synthesis and another O(N*(F+B)) for student training. The baseline methods take O(N*(Fg+Bg+F+B)) and O(Fg+F+B) for data synthesis and student training, respectively. We have omitted the number of iterations for each process for clarity.
> Let F,B denote time complexity for each forward and backward computation. In our method, data synthesis requires additional forward and backward computation for optimizing each batch of pixels. Using the same notations, the baseline methods take N*(Fg+Bg+F+B) where Fg and Bg denote forward and backward computation to train the generator. Each generator update requires forward and backward computation of the teacher model, hence (Fg+Bg+F+B). Then, (Fg+F+B) is required to generate the samples, and train the student model.
> We also report wall clock time in Appendix A.2, where our method takes 0.6 hrs on ResNet-20, 2.6 hrs on ResNet-56, and 3.6 hrs on WRN28-10. The compared methods (DaST, DFME, AIT, DFARD) take 1.5hrs using ResNet-20, 2.75hrs using ResNet-56, and 10hrs using WRN28-10 on RTX3090 under the same environment. The time is measured for generating 10000 synthetic samples on a single GPU.

---

> ### Author Response · Authors · 2023-11-15
> **Author Response to questions of Reviewer RMmq (2/2)**
>
> > **How to extract robust knowledge from clean images is not clearly expressed in this paper. Why can adversarial robustness be obtained using only loss constraints?**
>
> $\to$ In our method we do not extract robust knowledge from clean images. Instead, we extract knowledge from a pretrained teacher to generate synthetic samples, which are then used in adversarial training. To be specific, we use a pretrained network as guidance to optimize pixels from random Gaussian noise to image-like distributions. First, a set of random Gaussian noise is forwarded through the teacher network. Then, a set of optimization terms is computed using its output and/or intermediate features and statistics. Lastly, the computed loss is backpropagated all the way up to the input, where the gradient is used to update each pixel using Adam optimizer. Then, the generated samples are used in adversarial training.
>
> > **If this is the case, can it be used in any confrontation training?**
>
> $\to$ Yes. As shown in Table 6 of the main manuscript, the set of generated images can be used in any adversarial training methods that require a set of input images.  (Is ‘confrontation’ referring to adversarial? If not, please clarify the meaning of “confrontation”.) The table shows results for using different variations of adversarial training methods with our generated images.
>
>
> > **What is the relationship between adversarial robustness and the amount of generated data?**
>
> $\to$ As described in Appendix C, adding more generated training samples increases the robust accuracy, but saturates at some point (5-60,000 samples for CIFAR-10), where further increment in data size does not lead to meaningful performance boost. We have the sample quantity-accuracy plot in Appendix C for the suggested study. Thank you!

---

> > ### Comment · Reviewer_RMmq · 2023-11-22
> >
> > The authors' response resolves most of our concerns, but we still question the practical deployment of limited adversarial robustness. In addition, following Reviewer nfcV, recent DiffPure [1] can defend against adversarial attacks without training data.
> >
> > [1] Diffusion Models for Adversarial Purification, ICML 2022

---

> > > ### Author Response · Authors · 2023-11-22
> > > **Author Response to Reviewer RMmq**
> > >
> > > We thank the reviewer for further discussion on our work! We would like to address the remaining concern:
> > > > **Recent DiffPure can defend against adversarial attacks without training data.**
> > >
> > > $\to$ We would like to gently argue that DiffPure is not considered **data-free**, as it depends on a **pre-trained diffusion model on the same domain dataset**. The term data-free used in our context[1-7], not just assumes that the original train dataset is unavailable, but denies any alternative data of similar kind.
> > >
> > >   However, existing purifier/cleansing methods make a strong assumption that either 1) an abundant set of alternative dataset is available or 2) pretrained network (detector, purifier, diffusion, etc) trained on some massive dataset is available. This assumption is unrealistic in many applications including biomedical or biometric areas, where good quality data is scarce, or prohibited for privacy, proprietary reasons. Note that DiffPure is evaluated under general domain datasets, and its generalization to unseen domains have not been studied.
> > >
> > > We thank the reviewer for asking an important question, and giving us a chance to emphasize the necessity of our work!
> > >
> > >
> > > [1] Nayak, G.K. et al. "Zero-Shot Knowledge Distillation in Deep Networks. In International Conference on Machine Learning, 2019.
> > > [2] Chen, H. et al. DAFL:Data-Free Learning of Student Networks. In Proceedings of the IEEE International Conference on Computer Vision, 2019.
> > > [3] Nagel, M. et al. Data-free quantization through weight equalization and bias correction. In Proceedings of the IEEE/CVF International Conference on Computer Vision, 2019.
> > > [4] Cai, Y. et al. ZeroQ: A novel zero shot quantization framework. In Proceedings of the IEEE/CVF Conference on Computer Vision and Pattern Recognition, 2020.
> > > [5] Xu, S. et al. Generative low- bitwidth data free quantization. In European Conference on Computer Vision, 2020.
> > > [6] Li, Zhikai, Liping, Ma, Mengjuan, Chen, Junrui, Xiao, Qingyi, Gu. "Patch Similarity Aware Data-Free Quantization for Vision Transformers." European Conference on Computer Vision. 2022.
> > > [7] Truong, Jean-Baptiste, et al. "Data-free model extraction." Proceedings of the IEEE/CVF In conference on computer vision and pattern recognition. 2021.

---

### Author Response · Authors · 2023-11-20
**General Response to Reviewers**

We sincerely thank the reviewers for going through our work. We appreciate all the questions and comments, and found them to be constructive in improving our work.

We have uploaded the revised version of our manuscript, which includes all the additional experiments we have done during the rebuttal. We can summarize the changes as below:
 * Add test time defense methods for comparison (**F6PX, nfcV**) in Appendix.F.
 * Add evaluation under adaptive attacks (**F6PX**) in Appendix.G.

As the rebuttal phase is coming to an end, we await for the reviewers’ response and look forward to engaging in further discussions.

---

### Meta-Review · Area_Chair_zPW7 · 2023-12-05

**Metareview:**

This submission suggests data-free adversarial robustness as a problem setting. Given an existing, non-robust model, the submission uses model inversion approaches to generate synthetic data, which is then used for robust training. This strategy is evaluated on several general-purpose vision datasets, and on medical application datasets.

However, reviewers were ultimately not convinced by the arguments brought forth in this submission arguing that this is a sound strategy. The consensus is that the approach is limited in practical utility and the evaluation not sufficiently thorough given the claims in the paper. These issues center around the comparison to related approaches. While we understand that the authors describe a data-free approach, this should be contextualized by a clearer comparison to adjacent strategies, such as problems where public domain data is available, which has only happened in a limited manner during the rebuttal period. Further, the evaluation in this work would be strengthened by a strong evaluation of the proposed defense that does not consider adaptive attacks as an afterthought.

Finally, the privacy perspective appears flawed to me. The submission discusses a scenario where a model is available, but, due to apparent privacy reasons, its training data is not. The proposed approach then attempts to faithfully synthesize the training data from the model. To me, this amounts to a data loophole, and while this approach may be circumventing current regulations, I was not convinced that this is a promising strategy for the future, from reading this draft.

**Justification For Why Not Higher Score:**

The arguments brought forth in the submission in favor of the proposed setting were not convincing to reviewers. Practical utility was hence valued as limited and the provided experimentation was considered not sufficiently thorough.

**Justification For Why Not Lower Score:**

N/A

---

### Decision · Program_Chairs · 2024-01-16

Reject